# Exo-plore: Exploring Exoskeleton Control space through Human-Aligned Simulation

**Geonho Leem[1], Jaedong Lee[2], Jehee Lee[1], Seungmoon Song[3], Jungdam Won[1†]**
[1]Seoul National University, [2]Holiday Robotics, [3]Northeastern University

## Abstract

Exoskeletons show great promise for enhancing mobility, but providing appropriate assistance remains challenging due to the complexity of human adaptation to external forces. Current state-of-the-art approaches for optimizing exoskeleton controllers require extensive human experiments in which participants must walk for hours, creating a paradox: those who could benefit most from exoskeleton assistance, such as individuals with mobility impairments, are rarely able to participate in such demanding procedures. We present Exo-plore, a simulation framework that combines neuromechanical simulation with deep reinforcement learning to optimize hip exoskeleton assistance without requiring real human experiments. Exo-plore can (1) generate realistic gait data that captures human adaptation to assistive forces, (2) produce reliable optimization results despite the stochastic nature of human gait, and (3) generalize to pathological gaits, showing strong linear relationships between pathology severity and optimal assistance[1].

## 1 Introduction

Human mobility is fundamental to meeting basic life needs. While exoskeleton research shows promise for enhancing mobility (Molinaro et al., 2024; Slade et al., 2022), providing appropriate assistance, especially for users with atypical gait patterns, remains a challenge (Kim et al., 2024; Kang et al., 2025). Users alter their movement patterns and redistribute muscle coordination in response to assistance (Song & Collins, 2021), which can erode the benefits predicted by fixed-gait assumptions (Dembia et al., 2017). To personalize assistance, researchers have employed human-in-the-loop optimization (HILO) techniques, which constrain the exoskeleton controller design to a small set of tunable parameters (e.g., peak torque timing, magnitude, and duration) and systematically explore this parameter space through human experiments (Zhang et al., 2017). While HILO has proven effective for able-bodied young and older adults (Slade et al., 2022; Lakmazaheri et al., 2024), it requires demanding experiments where participants must walk for hours while wearing an exoskeleton. This situation presents a paradox: those who would benefit most from exoskeleton assistance, such as individuals with mobility impairments, are rarely able to participate in these extensive optimization procedures.

Neuromechanical simulations (Song et al., 2021) that couple musculoskeletal dynamics with control policies to predict how humans adapt to assistance have been proposed as a promising complement to exhaustive HILO (Luo et al., 2024; Firouzi et al., 2025; Tan et al., 2025). Yet policy design remains a bottleneck for reliable prediction. Biologically inspired controllers optimized with task cost functions can generate human-like gait in certain scenarios, but application and validation for assistive devices and pathological gaits remain limited (Song & Geyer, 2017; Geijtenbeek, 2019; Bersani et al., 2023). Deep reinforcement learning (Deep RL) offers an alternative that is less constrained by hand-specified control assumptions and potentially more generalizable, but most implementations that produce human-like motions rely on tracking reference motion-capture data to manage large observation and action spaces (Lee et al., 2019; Song et al., 2021; Park et al., 2025). Pure prediction is inherently challenging; practical approaches likely require both fitting controllers to existing data and predicting responses in extended scenarios Park et al. (2022). However, there is no unified framework to (i) fit controllers to observed human adaptations and (ii) predict responses in unobserved assistive conditions, particularly for exoskeleton assistance in gait pathology.

---

[†]Corresponding author (`jungdam@imo.snu.ac.kr`)
[1]Code and videos available at `https://daebangstn.github.io/exo-plore`.

In this paper, we present Exo-plore, a neuromechanical-simulation framework that discovers hip exoskeleton control parameters by coupling a Deep RL gait data generator with a stochasticity-aware surrogate optimizer. The gait data generator is tuned to fit human experimental trends by matching assistive moment/power scaling across assistance settings and walking speeds. This is achieved through a reward designed to align with empirical adaptation patterns, which combines metabolic energy minimization with a human–exoskeleton-interaction term based on resistance minimization hypothesis. As a result, the generator adapts across hip-assistance levels and musculoskeletal impairments, producing plausible assisted gaits in both healthy and pathological settings that align with empirical trends. We then train a surrogate network on abundant simulation data from the gait data generator to obtain smooth, differentiable CoT (cost of transport) landscapes to stabilize optimization under RL and simulation stochasticity. Using Exo-plore, we (i) reproduce assisted and unassisted gait trends consistent with human experiments, including assistive moment/power scaling and realistic metabolic reduction rates; (ii) identify speed-dependent optimal exoskeleton control parameters for able-bodied gait (optimal torque delay decreases as speed increases); and (iii) generalize to pathological gaits, revealing strong linear relations between pathology severity and optimal assistance in four of five conditions.

## 2 RELATED WORK

**Human-in-the-loop optimization (HILO) for exoskeletons.** Exoskeletons have demonstrated the ability to provide various benefits, including improved walking speed and reduced metabolic energy consumption (Molinaro et al., 2024; Slade et al., 2022; Lakmazaheri et al., 2024). However, their effectiveness varies significantly depending on control parameter settings and users' adaptation to assistance (Poggensee & Collins, 2021). HILO methods have been widely used to address both challenges by iteratively optimizing controllers based on real human responses and by evaluating candidate parameters using metrics such as metabolic cost of transport (CoT) after an adaptation (Zhang et al., 2017; Slade et al., 2024; Witte et al., 2020; Bryan et al., 2021; Song & Collins, 2021). In addition to CoT optimization, several studies have also optimized parameters based on user preference or perceived comfort (Ingraham et al., 2022; Lee et al., 2023). However, because experiments involved human participants, the number of iterations that can be performed to update parameters is often limited to fewer than 30 iterations, even when using auxiliary techniques such as online simulation (Gordon et al., 2022). Furthermore, individuals who might benefit most from exoskeleton assistance (e.g., mobility-impaired individuals) often show limited tolerance to this optimization process (Welker et al., 2021; Lakmazaheri et al., 2024), making the application of HILO often infeasible for these groups.

**Neuromechanical simulation.** Neuromechanical simulations provide physics-based digital twins for studying human movement, assistive devices, and neuromuscular impairments by coupling multibody musculoskeletal models with Hill-type muscle actuators and a control policy that drives muscle-driven dynamics (Song et al., 2021). Despite advances in modeling and simulation fidelity, predictive capability hinges on the control model, which remains only partially understood and difficult to specify. Existing approaches either (i) treat control as a black box to reproduce target motions, via trajectory optimization (De Groote & Falisse, 2021) or Deep RL (Song et al., 2021)—or (ii) rely on hand-crafted, biologically inspired controllers tuned for habitual gait (Song & Geyer, 2015; Bersani et al., 2023). These strategies can yield plausible motions but often face limits in generalization and computational efficiency, especially when predicting adaptations to novel exoskeleton assistance or pathological conditions.

**Deep RL-based Musculoskeletal Control.** Deep RL has shown promising results in neuromechanical simulation, but raises open questions about physiological fidelity and generalization (Lee et al., 2019). Recent studies use Deep RL within neuromechanical simulations to explore potential surgical effects, simulate gait impairments arising from altered muscle function, and synthesize diverse movements (Lee et al., 2019; Park et al., 2022; 2023; 2025). However, the sim-to-real gap remains largely unvalidated for muscle activations, metabolic energy, and adaptability to external assistance, which are critical for translational impact. While Luo et al. (2024) applied Deep RL to exoskeleton control, their formulation depends on imitation policies, limiting adaptation to unseen conditions. As a result, the simulation results do not align with natural human movement, and their correlation with real experimental outcomes has not been validated. In contrast, Exo-plore fits to established experimental trends and predicts responses in unseen conditions, enabling estimation of biomechanical metrics essential for exoskeleton applications.

## 3  METHOD: THE EXO-PLORE FRAMEWORK

Figure 1 presents an overview of our Exo-plore framework, which operates on a musculoskeletal character equipped with a hip exoskeleton (Lim et al., 2019a). The exoskeleton controller generates hip assistive torque $\tau_{\mathrm{exo}}(t)$ as follows:

$$\tau_{\mathrm{exo}}(t) = \kappa \cdot u(t - \Delta t) \tag{1}$$

where $u(t) = \sin(\theta_r) - \sin(\theta_l)$ is a control signal encoding relative motion of the two legs. $\theta_r$, $\theta_l$ denote the low-pass filtered right and left hip joint angles, respectively. The sine function bounds the joint angle signal to the range [-1, 1]. $\kappa$ and $\Delta t$ are control parameters, where the gain $\kappa$ scales the torque magnitude while the delay $\Delta t$ introduces temporal lag for the output (i.e., delayed-feedback control). Since the assistive torque is proportional to the relative motion between the two hip joints, $\kappa$ effectively acts as a stiffness parameter. This formulation therefore resembles a simplified impedance-control mechanism.

The objective of Exo-plore is to find the optimal control parameters $(\kappa, \Delta t)$ that minimize the simulated character's metabolic cost-of-transport during walking. This is achieved through two components: the gait data generator, which produces gait data that align with real human experiments, and the exoskeleton optimizer, which efficiently explores the control parameter space to identify the optimal parameters.

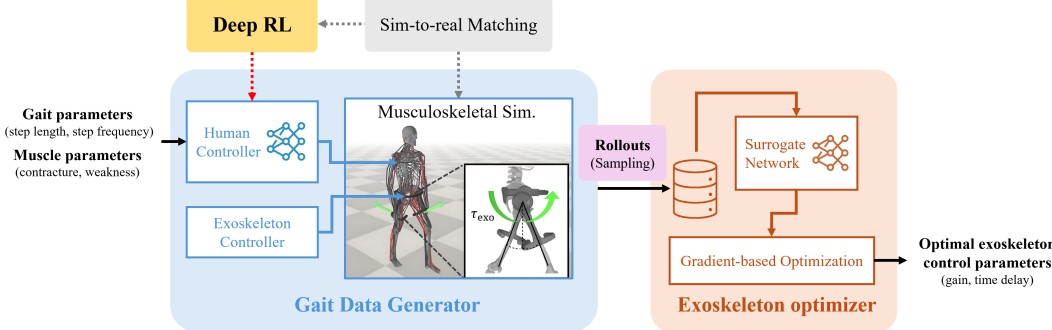

Figure 1: **The Exo-plore framework.** The generator produces gait trajectories under hip exoskeleton assistance using a musculoskeletal character with 164 muscles. Through our sim-to-real matching approach, these trajectories replicate the adaptation patterns and metabolic responses observed in real human experiments. The generated data are then used to train a surrogate network, which enables efficient and customizable optimization of exoskeleton control parameters.

### 3.1  GAIT DATA GENERATOR

The human controller in the gait data generator consists of three modules: a pose network (PoseNet), which computes positional-derivative (PD) target joint positions $\mathbf{q}_d$; a PD controller that generates joint torques $\boldsymbol{\tau}_{\mathrm{PD}}$ to reduce the error between the target and current joint positions; then compensate the exoskeleton assistance $\tau_{\mathrm{exo}}$ from $\boldsymbol{\tau}_{\mathrm{PD}}$ to compute the torque to be generated by the muscles $\boldsymbol{\tau}_{\mathrm{MCN}}$; and a muscle coordination network (MCN), which maps the computed target joint torques $\boldsymbol{\tau}_{\mathrm{MCN}}$ to muscle activations $\mathbf{a}$. These activations are then applied to the underlying musculoskeletal dynamics, and the states of the musculoskeletal character and the exoskeleton are updated accordingly. The PoseNet and MCN are jointly trained during the iterative simulation process using Deep RL and supervised learning, respectively, while a stable PD is used for our PD control (Tan et al., 2011). The detailed architecture of the human controller is provided in Appendix A.

To train the PoseNet, we follow the formulation of Generative GaitNet (Park et al., 2022), but we extend the state, action, and reward to incorporate the exoskeleton and improve motion naturalness. Here, we describe only the crucial components, including our extensions, while other technical details are provided in Appendix B. The state is defined as $\mathbf{s} = (s_{\mathrm{skeleton}}, s_{\mathrm{muscle}}, s_{\mathrm{exo}})$, where $s_{\mathrm{skeleton}}$ and $s_{\mathrm{muscle}}$ represent the physical states of the musculoskeletal character, and $s_{\mathrm{exo}}$ denotes the exoskeleton state. The total reward is defined as

$$r_{\mathrm{total}} = w_{\mathrm{gait}} \cdot r_{\mathrm{gait}} + w_{\mathrm{arm}} \cdot r_{\mathrm{arm}} + w_{\mathrm{energy}} \cdot r_{\mathrm{energy}} + w_{\mathrm{HEI}} \cdot r_{\mathrm{HEI}} \tag{2}$$

where $r_{\text{gait}}$ encourages the character to follow the given target gait, and is defined as the product of four components as $r_{\text{gait}} = r_{\text{step}} \cdot r_{\text{vel}} \cdot r_{\text{head}} \cdot r_{\text{sway}}$, where $r_{\text{step}}$ encourages the specified step length, $r_{\text{vel}}$ encourages the specified walking speed, and $r_{\text{head}}$ and $r_{\text{sway}}$ stabilize the head and body sway, respectively. To prevent unnatural arm movements, an imitation reward $r_{\text{arm}}$ is introduced. The energy reward, $r_{energy} = 1 - k_{energy} \cdot \text{MEE}$, regularizes energy expenditure during walking, where MEE denotes the metabolic energy expenditure, computed as follows:

$$\frac{d}{dt}\text{MEE}(\mathbf{a};\ \alpha,\ \beta) = \sum_i m_i^\alpha a_i^\beta \tag{3}$$

where $m_i$ and $a_i$ are the mass and activation of the $i$-th muscle, respectively, and $\alpha$, $\beta$ are tunable design parameters. This allows flexible adjustment between effort-like ($\alpha$-dominant) and fatigue-like ($\beta$-dominant) behaviors, in line with previous studies (Ackermann & Van den Bogert, 2010). These are crucial parameters that must be properly adjusted to ensure that the simulation results align with those from real human experiments. The detailed design of $\alpha$ and $\beta$ is presented in Section 3.2.

To account for human adaptation during exoskeletal assistance, we introduce a novel reward $r_{\text{HEI}}$ that explicitly models physical human-exoskeleton interaction (HEI). In our framework, $r_{\text{HEI}}$ plays a critical role in bridging the sim-to-real gap under exoskeleton assistance. This reward is also further discussed in Section 3.2.

The MCN is trained in a supervised manner to learn the mapping from desired joint torque $\boldsymbol{\tau}_{\text{d}}$ to muscle activations $\mathbf{a}$. The total loss is computed as

$$\mathcal{L}_{\text{MCN}} = \left\| \boldsymbol{\tau}_{\text{MCN}} - \boldsymbol{\tau}(\mathbf{a}) \right\|^2 + w_{\text{reg}} \left\| \mathbf{a} \right\|^2 + w_{\text{IMR}}\, \mathcal{L}_{\text{IMR}}. \tag{4}$$

The first term guides the generated torque $\boldsymbol{\tau}(\mathbf{a})$ from the output muscle activations $\mathbf{a}$ to match the desired torque $\boldsymbol{\tau}_{\text{MCN}}$, while the second term penalizes excessively large activations. Please refer to (Lee et al., 2019) for more technical details. Large muscles such as the gluteus medius and soleus are represented by multiple line muscles. Controlling each line muscle independently could lead to physiologically implausible results. To prevent this, we introduce the final term, the intra-muscular regularizer (IMR), which enforces coherent activations among the line muscles belonging to the same anatomical muscle group. Specifically, the IMR penalizes deviations from the mean group activation, which is computed as follows:

$$\mathcal{L}_{\text{IMR}} = \sum_{g=1}^{N_g} \sum_{j \in \mathcal{G}_g} \mathbf{1}\big(|a_j - \bar{a}_g| > 0.1\big) \big(a_j - \bar{a}_g\big)^2, \qquad \bar{a}_g = \frac{1}{|\mathcal{G}_g|} \sum_{j \in \mathcal{G}_g} a_j. \tag{5}$$

where $a_j$ denotes the activation of the $j$-th muscle in muscle group $g$, and $\bar{a}_g$ is its corresponding mean group activation. The number of muscle groups and the set of indices of the line muscles in muscle group $g$ are denoted by $N_g$ and $\mathcal{G}_g$, respectively. For details on $\mathcal{L}_{\text{IMR}}$, see Appendix C.

## 3.2 Sim-to-real Matching in Gait Data Generator

The simulation results generated by the Gait data generator need to align with real human experimental results to ensure that the control parameters optimized in simulation are applicable to real-world scenarios, a process often referred as bridging the sim-to-real gap. To this end, we focus on reproducing two core human behaviors observed in several biomechanics studies: the principle of energy minimization and the active adaptation to exoskeleton assistance.

First, many studies have observed that human walking tends toward an energetically optimal gait (Umberger & Martin, 2007; Zarrugh et al., 1974). This means that humans adjust their step size and step frequency at each walking speed to minimize the metabolic cost-of-transport (CoT), which is the metabolic energy expenditure per unit travel distance. As reported in previous studies (Maxwell Donelan et al., 2001), measured CoT values across different walking speeds form an upward-opening parabola, and humans tend to select the minimum point as their preferred walking speed (PWS) under nominal conditions. Based on this, we determine the MEE parameters $(\alpha, \beta)$ in Equation 3 so that the resulting *Walking speed - CoT* curve and the corresponding PWS best match those reported in real experiments. For this, we develop two algorithms (Algorithm 1 and 2). The first algorithm assumes feasible values of $(\alpha, \beta)$ (Line 2), trains a gait data generator to compute the PWS (Line 4), and selects the optimal values of $(\alpha, \beta)$ that best match real human PWS (Line 5-8). The second algorithm performs rollouts of gait trajectories across feasible parameters (Line

2-3), calculates the CoT for each, and determines the PWS that yields the minimum CoT (Line 4-7). We define two procedures used in algorithm 1, 2 and 3.

**trainGenerator.** This procedure trains a human controller conditioned on a given metabolic energy model $E_m$:

$$G = \texttt{trainGenerator}(E_m). \tag{6}$$

During this training process, the simulated musculoskeletal character does not wear an exoskeleton, as we evaluate PWS in the absence of external assistance. Consequently, PoseNet does not use the exoskeleton-related input $r_{\text{HEI}}$.

**rollout.** This procedure generates a gait trajectory from the trained gait data generator $G$ for given gait parameters $(L, f)$:

$$\tau_{\text{cyc}} = \texttt{rollout}(G(L, f)). \tag{7}$$

To ensure stable trajectory generation, we discard the first five gait cycles after the simulation starts and use the subsequent ten cycles to construct the trajectory.

Based on those algorithms, the optimal parameters for our framework were determined as $(\alpha, \beta) = (1.5, 1.0)$. A detailed technical discussion is provided in the Appendix D

---

**Algorithm 1** Adjusting the MEE to make Explore to predict human PWS.

**Notation:** muscle i mass, activation $m_i, a_i$

**Require:** Human PWS $v_{\text{real}}$
**Ensure:** MEE parameter $(\alpha^*, \beta^*)$
1: $e_{\min} \leftarrow +\infty$, $(\alpha^*, \beta^*) \leftarrow \varnothing$
2: **for** $(\alpha, \beta) \in$ candidate **do**
3:      $E_m \leftarrow \text{MEE}(\alpha, \beta)$
4:      $G_i \leftarrow \texttt{trainGenerator}(E_m)$
5:      $v_{\text{sim}} \leftarrow \texttt{evalPWS}(G_i, E_m)$
6:      $e \leftarrow |v_{\text{real}} - v_{\text{sim}}|$
7:      **if** $e < e_{\min}$ **then**
8:          $e_{\min} \leftarrow e$, $(\alpha^*, \beta^*) \leftarrow (\alpha, \beta)$
9:      **end if**
10: **end for**

**Algorithm 2** (evalPWS) evaluating PWS for gait data generator

**Notation:** gait step length $L$, gait step frequency $f$, gait cycle trajectory $\tau_{\text{cyc}}$

**Require:** Data generator $G$, metabolic energy expenditure $E_m$
**Ensure:** $v_{\text{sim}}$ (PWS of $G$)
1: $\text{CoT} \leftarrow +\infty$, $v_{\text{sim}} \leftarrow \varnothing$
2: **for** $(L, f) \in$ available **do**
3:      $\tau_{\text{cyc}} \leftarrow \texttt{rollout}(G(L, f))$
4:      $\text{CoT} = \frac{1}{L} \int_{\tau_{\text{cyc}}} dE_m$
5:      **if** $\text{CoT} < \text{CoT}_{\min}$ **then**
6:          $\text{CoT}_{\min} \leftarrow \text{CoT}$, $v_{\text{sim}} \leftarrow (L \times f)$
7:      **end if**
8: **end for**

---

Second, experimental studies suggest that humans do not simply adapt to the assistance torque (Normand et al., 2023; Livolsi et al., 2025). For example, at the same walking speed, increasing the delay parameter ($\Delta t$) from 0.15 s to 0.25 s results in a 1.5-fold increase in RMS assistive torque (Figure 6). The RMS assistive torque is proportional to the hip joint angle (Equation 1). This indicates that humans actively modify their kinematics, such as increasing step size, to further utilize the assistance.

To reproduce this adaptive behavior, we develop the human–exoskeleton interaction (HEI) reward based on a *resistance-minimization* hypothesis, which reflects the human tendency to avoid mechanical power loss. This is inspired by the behavioral principle of loss aversion, which states that people are more sensitive to losses than to equivalent profits (Kahneman, 1979; Tversky & Kahneman, 1991; Brown et al., 2024). Our HEI reward is formulated as follows:

$$r_{\text{HEI}} = 1 + \frac{1}{\kappa} \sum_{k \in \{L, R\}} \min(0, P_k) \tag{8}$$

where $P_k$ denotes the mechanical power applied by the exoskeleton on either the left or right side, and $\kappa$ is the gain parameter of the exoskeleton controller. Equation 8 takes effect when the exoskeleton applies resistive power to the human ($P_k < 0$), causing $r_{\text{HEI}}$ to drop below 1.

## 3.3 EXOSKELETON OPTIMIZER

The goal of the exoskeleton optimizer is to identify exoskeleton control parameters that minimize CoT, meaning the parameters that provide the most effective assistance to humans wearing the exoskeleton. Because the mapping between the control parameters and their resulting CoT is often highly non-linear, previous methods (HILO) typically relies on gradient-free and sampling-based optimization methods such as CMA-ES or Bayesian optimization (BO) (Slade et al., 2024; Gordon et al., 2022). These methods are particularly effective in HILO settings, where data collection is limited and expensive. For instance, BO often employs a Gaussian process (GP) as a surrogate model to approximate the CoT and guide the search more efficiently. However, in our simulation-based framework, where data can be generated as needed with low cost, the assumptions underlying such sample-efficient methods no longer apply. While GP offer excellent performance in low-data regimes, their poor scalability in both time and memory complexity makes them unsuitable for our framework (see Appendix E). To take advantage of the data-rich simulation environment, we instead use a neural network surrogate based on a multilayer perceptron (MLP). Neural networks scale efficiently with data and support fast, gradient-based optimization, making them a better fit for Exo-plore.

Surrogate networks trained on a naive uniform grid-sampled parameter set may exhibit artifacts due to the regularity of the sampling patterns. In particular, grid-based sampling may lead to aliasing effects and lacks the stochastic variation. To mitigate these issues, we instead use Latin hypercube sampling (LHS) (Forrester et al., 2008), which stratifies each parameter range and draws one sample per interval. This stratification reduces aliasing and ensures uniform coverage without the clustering of random sampling. Please refer to Appendix F for ablation studies.

Algorithm 3 outlines the overall process of optimizing the control parameters: For the simulation parameter space $\mathcal{X}$ used in training the gait data generator, we perform three steps: sampling candidate parameters and their evaluation

---

**Algorithm 3** Optimizing exoskeleton control parameters

**Require:** Simulation parameter space $\mathcal{X}$, Data generator $G$, Target gait parameter $g$
**Ensure:** Optimal control parameter $\mathbf{c}^*$
1: $\mathcal{B} \leftarrow \varnothing$
2: **for** $x_i \in \mathrm{LHS}(\mathcal{X})$ **do**
3: $\quad \tau_{\mathrm{cyc}} \leftarrow \mathtt{rollout}(G(L, f))$
4: $\quad y \leftarrow \frac{1}{L} \int_{\tau_{\mathrm{cyc}}} d\mathrm{E}_m$
5: $\quad \mathcal{B} \leftarrow \mathcal{B} \cup \{(x_i, y)\}$
6: **end for**
7: $\hat{f}_\theta(\mathbf{c}) \leftarrow \mathrm{fit}(\mathcal{B})$
8: $\mathbf{c}^* \leftarrow \arg\min_{\mathbf{c}} \hat{f}_\theta(\mathbf{c}; g)$
9: $\qquad\qquad\quad \triangleright$ gradient-based optimization

---

through simulation (Line 2-5), training a surrogate network (Line 6), and performing gradient-based optimization to minimize CoT given a target gait (Line 7).

To train the surrogate network, we minimize the following loss:

$$\mathcal{L}_{\mathrm{surrogate}} = \mathcal{L}_{\delta_h}(\hat{y}, y) + \lambda_{\mathrm{grad}} \|\nabla_x \hat{y}\|_2^2 + \lambda_{\mathrm{L1}} \sum_i |w_i| + \lambda_{\mathrm{L2}} \sum_i w_i^2. \qquad (9)$$

where $\hat{y}$ is the predicted CoT value from the surrogate network, $y$ is the ground truth value obtained from simulation. We used Huber loss ($\mathcal{L}_{\delta_h}$) (Hastie et al., 2009) to reduce the effect of outliers that may arise from the instability and stochasticity of the simulation. The gradient penalty ($\nabla_x \hat{y}$) encourages smoothness of the surrogate landscape by constraining the norm of the output gradient with respect to its input (Appendix G). In addition, the L1 and L2 weight penalties act as standard regularizers. This formulation enables the trained surrogate network to generalize well in predicting CoT values for unseen control parameters while ensuring stable gradient behavior. For gradient-based optimization, we use sequential least-squares quadratic programming (SLSQP) and trust-region algorithms. Please refer to Appendix H for the optimization objective.

## 4 RESULTS

### 4.1 SIMULATION SETUP

**Simulation model.** We use the musculoskeletal model from prior work (Lee et al., 2019; Park et al., 2022), which consists of 23 bones and 164 muscles. The bones are simulated as rigid bodies, and the muscles are modeled as Hill-type muscle actuators (Hill, 1938). The simulation is performed

in the DART physics engine (Lee et al., 2018). In this model, tendons are represented as rigid rather than compliant to reduce the computational burden induced by Deep RL. To further reduce computational cost, we apply muscle actuation to the lower body only, while torque-based actuation is used for the upper body. For more technical details about simulated model, see Appendix I.

**Generalization to pathological gait.** We model five pathological gait conditions (calcaneal gait, foot drop, equinus gait, crouch gait, and waddling gait) by implementing specific alterations to muscle strength and contracture parameters. We trained separate Exo-plore models for each condition, rather than employing a unified mixture-of-experts approach (Park et al., 2022). To enhance generalization, we randomized simulation parameters during human controller training. Technical details are provided in Appendix J.

## 4.2 GAIT GENERATION – UNASSISTED

We first validate whether our gait data generator can produce results that align with real human experiments under the *unassisted* setup (i.e., $\kappa = 0$). This serves as a useful sanity check, since several human experiments without assistance have been previously reported.

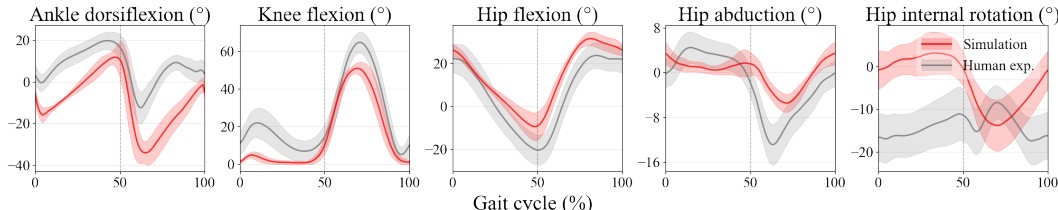

Figure 2: Comparison of joint kinematics over a gait cycle, with simulated results shown in red and human experimental data in gray.

Figure 2 shows a comparison of joint kinematics between our simulated results and human experiments (Boo et al., 2025), where the joint angles of the ankle, knee, and hip are reported. To obtain the simulated results, we trained the gait data generator with multiple random seeds and averaged the resulting joint angles over one gait cycle. Overall, joint kinematics qualitatively match with human data, except for hip internal rotation. We suspect that this discrepancy occurred due to overfitting of the Deep RL policy to the specific training environment, and thus it could be improved by modifying training conditions, such as changing the terrain of the environment. The kinematics of the pathological gaits are shown in Appendix K.

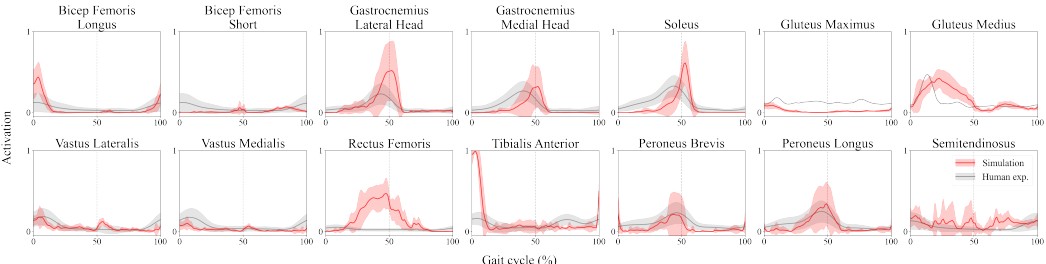

Figure 3: Comparison of simulated and experimental muscle activations.

Figure 3 compares simulated and experimental muscle activations (Boo et al., 2025) over one gait cycle. The results show that similar activation patterns to those observed in humans can be generated, even though we did not explicitly incorporate human muscle activation patterns into the training process. Note that activation timing and patterns are more informative than absolute values because muscle activations in human data (i.e., EMG signals) are normalized to each muscle's maximum voluntary contraction, which can vary widely with setup. For more discussion, please refer to Appendix L. In addition, we performed a detailed analysis of the ground reaction forces (GRF). Additional information can be found in the Appendix M.

Additionally, we demonstrate the importance of choosing appropriate metabolic energy expenditure (MEE, Equation 3) parameters in Figure 4. The leftmost figure shows the *Walking speed – CoT* curves obtained with different parameter choices. The curve with the optimal parameters chosen by Algorithm 1 best matches the human experiment results (Browning et al., 2006) in terms of overall trend and PWS. The middle and rightmost figures show the predicted gait parameters (step length and step frequency) as walking speed increases. The prediction results from our gait data generator match reasonably well with human experiments (Schwartz et al., 2008), except for the increased step frequency at low walking speeds

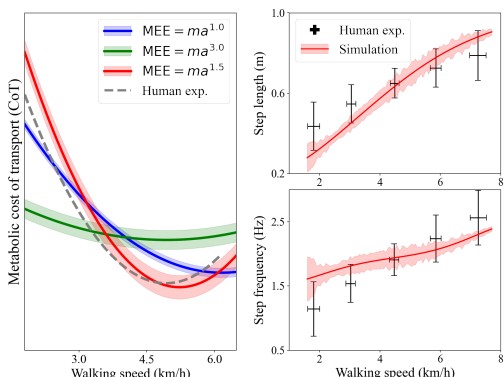

Figure 4: Walking speed vs (CoT, Step length, Step frequency)

($< 4\,\mathrm{km/h}$). This discrepancy might be because the Deep RL policy often had difficulty producing gaits with short step lengths due to the simplified foot model (box-shaped and rigid), which often caused unwanted collisions.

## 4.3 Gait Generation – Assisted

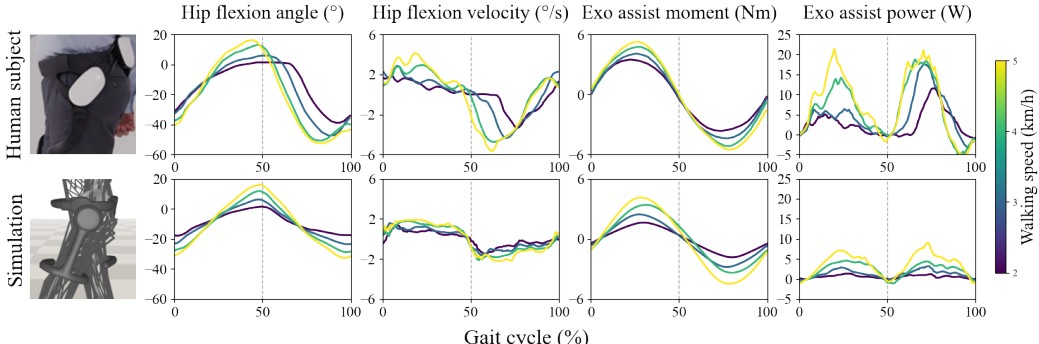

Figure 5: Comparison of gait kinematics and dynamics under exoskeleton assistance between human experimental results (top) and simulated results (bottom).

Figure 5 compares our simulation results with human experiments (Lim et al., 2019b) under exoskeleton assistance with the control parameters $(\kappa, \Delta t) = (8\,\mathrm{Nm}, 0.25\,\mathrm{s})$. Two kinematic results (hip flexion angle and its velocity) and two dynamic results (assistive moment and power) are reported, and the overall trends match reasonably across different walking speeds, demonstrating the effectiveness of our framework. Although overall trends are consistent across all four results, gaps in absolute values may result from discrepancies in musculoskeletal conditions between the simulated character and human participants, as well as from potential overfitting of the Deep RL policy to the fixed environment.

We incorporated the human-exoskeleton interaction (HEI) reward to better model human adaptation to exoskeleton assistance. To validate its effectiveness, we conducted an ablation study and a comparison with an alternative design choice like $r_{\mathrm{assist}}$ or no $r_{\mathrm{HEI}}$ (Appendix N). Figure 6 reports RMS assistive moment, mean assistive power, and mean resistive power across different walking speeds. In the human experiment (Lim et al., 2019b), the results resemble an increasing line, a downward-opening parabola, and an upward-opening parabola, respectively. $r_{\mathrm{HEI}}$ is our original formulation based on resistance minimization, no $r_{\mathrm{HEI}}$ means no HEI reward is incorporated, and $r_{\mathrm{assist}}$ means that assistance maximization is used. The results with our HEI reward show similar trends to the human experiments and also yield the highest Pearson correlation with them. For example, at $4\,\mathrm{km/h}$, human experiments show a 1.88-fold increase in assistive power when the delay increases from $0.05\,\mathrm{s}$ to $0.25\,\mathrm{s}$. With $r_{\mathrm{HEI}}$, the result showed a similar 1.73-fold increase (correlation = 0.83), whereas no $r_{\mathrm{HEI}}$ and $r_{\mathrm{assist}}$ showed weaker or inconsistent scaling (0.67- and 1.04-fold; correlation = 0.69 and 0.23, respectively). Figure 7 shows the maximum metabolic reduction rate

achieved through exoskeleton assistance. Our HEI reward achieved the closest alignment with human experimental results (Lim et al., 2019b), whereas the others produced lower metabolic reduction rate values, indicating that the simulated characters did not realize the same level of benefit observed in human participants. Evaluated metabolic reduction rate in the simulation is slightly higher than in human experimental data. Note that simulation metabolic reduction rate is computed relative to the unassisted condition ($\kappa = 0$), whereas human experiments measure it relative to walking without an exoskeleton, partly explaining the discrepancy. Additional details are provided in Appendix O.

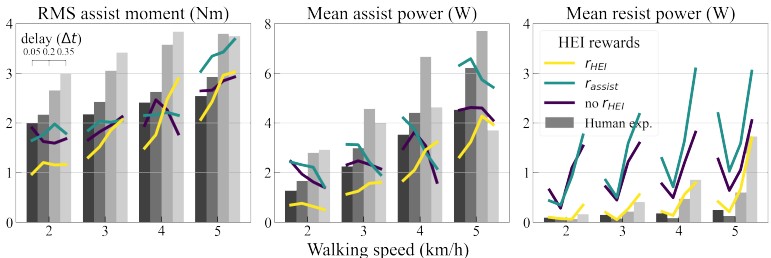

Figure 6: Comparison of assistive moment, assistive power, and resistive power across walking speeds, showing that our HEI reward ($r_{\text{HEI}}$) reproduces the trends observed in human experiments and achieves the highest correlation with empirical data.

Figure 7: Comparison of maximum metabolic reduction rate.

## 4.4 OPTIMIZING EXOSKELETON CONTROL PARAMETERS

We demonstrate the applicability of Exo-plore by discovering and analyzing optimal exoskeleton control parameters for both able-bodied and disabled individuals.

**Able-bodied Individuals.** To optimize exoskeleton control parameters, we train a surrogate network to learn the mapping between CoT (i.e., metabolic energy consumption per unit travel distance) and the control parameters ($\kappa$: gain, $\Delta t$: time delay). Figure 8(a-c) show the raw training data obtained from simulation and the learned mappings with two gradient penalty settings ($\lambda_{\text{gp}} = 0.1$ and $\lambda_{\text{gp}} = 0.01$), respectively. While the raw data contain many local minima from simulation stochasticity and Deep RL training, the surrogate network provides a smoother landscape that enables faster and more stable optimization while still capturing essential trends. A trade-off exists between small and large gradient penalties: smaller penalties allow more aggressive optimization but risk overfitting to artifacts, whereas larger penalties reduce this risk but may limit sensitivity. Figure 8(d) shows the optimal control parameters obtained from the two surrogate networks. The optimized delay ($\Delta t$) decreases monotonically as speed increases.

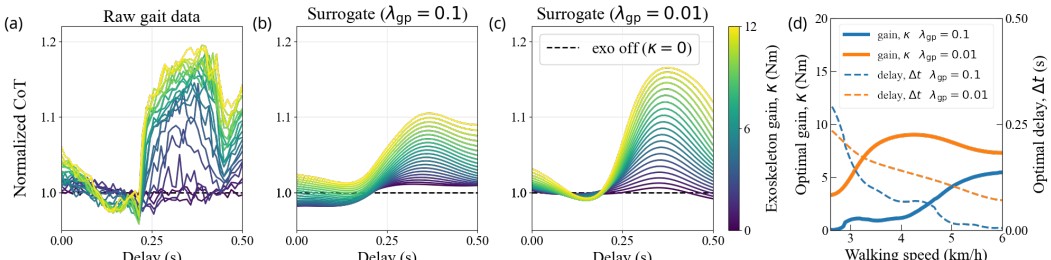

Figure 8: CoT landscape from raw simulation data (a), surrogate network trained with $\lambda_{\text{gp}} = 0.1$ (b), surrogate network trained with $\lambda_{\text{gp}} = 0.01$ (c), and optimized exoskeleton control parameters ($\kappa$, left axis; $\Delta t$, right axis) across walking speeds (d).

**Disabled Individuals.** We evaluate our framework on musculoskeletal impairment profiles representative of pathological gait. Pathology severity is quantified as the degree of deviation in muscle parameters from healthy reference values. We examine five common pathological gait types: equinus, waddling, crouch, calcaneus, and foot drop. Figure 9 presents the optimal gains ($\kappa$) identified by Exo-plore, averaged across walking speeds from 2 to 3.5 km/h, with values reported for different levels of pathology severity in each gait type. The optimal gains exhibit strong linear correlations with severity across all conditions except foot drop.

These correlations can be explained by the characteristics of each pathology. In equinus and waddling gaits, characterized by toe-walking and lateral trunk sway, respectively, the pathological patterns create greater instability compared to normal gait. External forces, such as exoskeleton assistance, may amplify this instability, necessitating minimal and carefully controlled intervention. In contrast, crouch and calcaneal gaits, characterized by excessive knee flexion and reduced ankle plantarflexion, respectively, present stable but metabolically demanding kinematic patterns where exoskeleton assistance effectively reduces energy expenditure without compromising stability. Foot drop exhibited irregular trends causing frequent toe-ground collisions, which substantially increased gait variability and prevented stable optimization convergence. Further detailed analysis is provided in Appendix P.

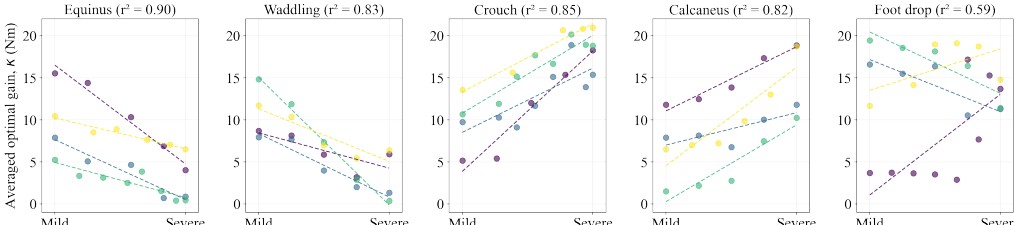

Figure 9: Optimal exoskeleton gain over five pathological gait types, with values reported for different levels of pathology severity in each gait type. Each color corresponds to different random seed. Detailed numerical results are provided in Appendix P.4.

## 5 CONCLUSION AND LIMITATIONS

**Conclusion.** This study developed Exo-plore, a simulation-based framework that optimizes exoskeleton controllers through neuromechanical simulation and Deep RL. Through sim-to-real matching, we demonstrated that Exo-plore can serve as an alternative to experimental trials for controller optimization. Furthermore, by extending the framework to pathological gaits, Exo-plore enables the design of exoskeleton controller for mobility-impaired individuals who are not suitable for traditional human-in-the-loop optimization methods.

**Limitations.** Despite these promising results, key limitations remain. Most importantly, experimental validation with human subjects, including patient populations, is required to establish the validity of simulation-based predictions and the clinical effectiveness of the optimized controllers. Within our framework, several components warrant evidence of real-world impact, including simplified reward models and the current lack of personalization to subject-specific motor control, and the use of approximate muscle dynamics (see Appendix R for details) that may fail to capture subject-specific neuromuscular responses to assistance.

### ACKNOWLEDGMENTS

This work was supported by the National Research Foundation (NRF) of Korea grant funded by the Korea government (MSIT) (RS-2024-00450647). This work was also supported by the Institute of Information & Communications Technology Planning & Evaluation (IITP) grant funded by MSIT under Grant Nos. RS-2025-25442338 (AI Star Fellowship Support Program, Seoul National University), RS-2021-II211343 (Artificial Intelligence Graduate School Program, Seoul National University), and IITP-2026-RS-2020-II201460 (ITRC: Information Technology Research Center).

### REPRODUCIBILITY STATEMENT

Code and videos available at `https://daebangstn.github.io/exo-plore`. The experimental data used in this work are properly cited in their respective figures.

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

## A  ARCHITECTURE OF HUMAN CONTROLLER

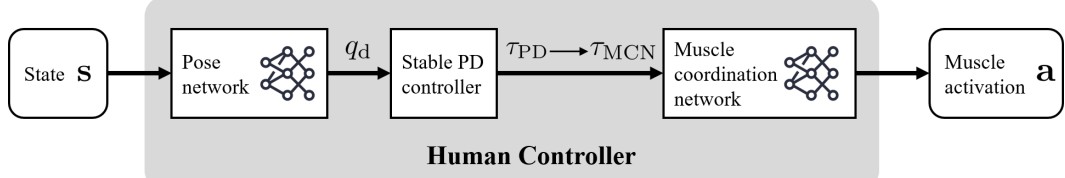

Figure A.1: The human controller computes muscle activations from the musculoskeletal simulation state and applies them back to the simulation.

## B  DETAILS OF THE RL FORMULATION FOR POSENET

### B.1  STATE REPRESENTATION

- **Kinematics** ($\mathbb{R}^{349}$)**:** All simulated rigid bodies are represented in a root-centered local coordinate frame, including their relative positions, orientations, and linear and angular velocities. Phase-based mirroring is applied to account for gait symmetry (Abdolhosseini et al., 2019).
    - Root height: $\mathbb{R}^1$
    - Root linear velocities: $\mathbb{R}^3$
    - Body part positions: $\mathbb{R}^{69}$
    - Body part rotations: $\mathbb{R}^{138}$
    - Body part linear velocities: $\mathbb{R}^{69}$
    - Body part angular velocities: $\mathbb{R}^{69}$
- **Gait features** ($\mathbb{R}^9$)**:**
    - Target foot position: $\mathbb{R}^3$
    - Foot contact state: $\mathbb{R}^2$
    - Target gait velocity: $\mathbb{R}^1$
    - Current/Target/Reference gait phase: $\mathbb{R}^3$
- **Muscle-related terms** ($\mathbb{R}^{59}$)**:** Passive muscle forces, lower and upper torque bounds per joint, and dynamic muscle-specific parameters like muscle length.
    - Muscle weakness and contracture: $\mathbb{R}^5$
    - Passive forces: $\mathbb{R}^{18}$
    - Torque bounds: $\mathbb{R}^{36}$
- **Exoskeleton assist torque history:** ($\mathbb{R}^{64}$)**:** Recent history of assistive torques applied at the hip joints (left/right), representing the temporal profile of exoskeletal assistance to the human.

### B.2  ACTION REPRESENTATION

As in Park et al. (2022), we employ the residual pose action formulation. The action consists of pose and phase modulation:

$$\mathbf{A} = (\Delta\phi, \Delta M), \tag{10}$$

where $\Delta M$ denotes the pose displacement, and $\Delta\phi$ represents the phase increment that either advances or delays gait phase with respect to the reference gait motion.

The gait phase($\phi$) and target pose($M(\phi)$) for the PD controller is then updated according to

$$\begin{aligned} \phi_{t+1} &= \phi_t + \Delta\phi, \\ M_t &= M_{\mathrm{ref}} \oplus \Delta M. \end{aligned} \tag{11}$$

where $M_{\mathrm{ref}}$ denotes the reference motion (obtained from Park et al. (2022)) and $\oplus$ denotes the operation between a pose and a displacement, which includes operations in SO(3) to properly update 3D rotations. For more technical details, refer to Lee (2008)

## B.3 REWARD DESIGN

We use a composite reward consisting of four terms:

$$r_{\text{total}} = w_{\text{gait}} \cdot r_{\text{gait}} + w_{\text{energy}} \cdot r_{\text{energy}} + w_{\text{arm}} \cdot r_{\text{arm}} + w_{\text{HEI}} \cdot r_{\text{HEI}} \tag{12}$$

Since we previously mentioned $r_{\text{energy}}$ and $r_{\text{HEI}}$ we now discuss the remaining one. There are four terms in the $r_{\text{gait}} = r_{\text{step}} \cdot r_{\text{vel}} \cdot r_{\text{head}} \cdot r_{\text{sway}}$. Specifically, $r_{\text{step}}$ and $r_{\text{vel}}$, encourage the agent to maintain a target step size and walking velocity:

$$r_{\text{step}} = \exp\left(-\left(\|\mathbf{h} - \mathbf{h}_{desired}\|/\sigma_{\text{step}}\right)^2\right),$$

$$r_{\text{vel}} = \exp\left(-\left(\|\mathbf{v} - \mathbf{v}_{desired}\|/\sigma_{\text{vel}}\right)^2\right). \tag{13}$$

Here, $\mathbf{h}$ denotes the current foot position, and $\mathbf{v}$ denotes the average forward walking velocity of the simulated character. Both values are compared against their respective targets to guide the agent toward producing the desired gait pattern.

Humans generally maintain head stability during walking, as key sensory systems like vision and balance are located there. $r_{\text{head}}$ encourages consistent head motion by penalizing abrupt changes in head angle, linear acceleration, and angular acceleration.

$$r_{\text{head}} = f_r \cdot f_v \cdot f_\omega,$$

$$f_r = k_{\text{alive}} + (1 - k_{\text{alive}}) \exp\left(-\left(\theta_{\text{head}}/\sigma_{\text{head}}\right)^2\right), \quad \text{if } \theta_{\text{head}} > \lambda_r, \text{ else } 1,$$

$$f_v = k_{\text{alive}} + (1 - k_{\text{alive}}) \exp\left(-\left(a_{\text{lin}}/\sigma_{\text{head}}\right)^2\right), \quad \text{if } a_{\text{lin}} > \lambda_v, \text{ else } 1,$$

$$f_\omega = k_{\text{alive}} + (1 - k_{\text{alive}}) \exp\left(-\left(a_{\text{ang}}/\sigma_{\text{head}}\right)^2\right), \quad \text{if } a_{\text{ang}} > \lambda_\omega, \text{ else } 1. \tag{14}$$

Here, $\theta_{\text{head}}$ denotes the head orientation with respect to the frame aligned to the progression direction and the global up direction. $a_{\text{lin}}$ and $a_{\text{ang}}$ represent the linear and angular accelerations of the head, respectively.

To prevent the emergence of unrealistic gait patterns due to model inaccuracies or excessive energy minimization, a sway reward discourages abnormal deviations in musculoskeletal body orientations, promoting kinematics that remain close to typical human walking behavior.

$$r_{\text{sway},b} = \exp\left(-\left(\frac{|q_b - \bar{q}_b| - \delta_b}{\sigma_{\text{sway}}}\right)^2\right) \quad \text{if } |q_b - \bar{q}_b| \leq \delta_b, \text{ 1 otherwise,}$$

$$r_{\text{sway}} = \prod_{b \in \mathcal{B}} r_{\text{sway},b}, \quad B = \{\text{pelvis, spine, foot}\}. \tag{15}$$

where $q_b$ denotes the angle of body $b$, $\bar{q}_b$ is the angle derived from normative gait data, and $\delta_b$ represents the deviation threshold for body $b$. The total sway reward $R_{\text{sway}}$ is computed as the product of body-wise sway terms $r_{\text{sway},b}$ across joints.

To promote natural arm movements during gait, we introduce an upper-arm imitation reward. This reward measures the deviation between the simulated and reference upper-arm joint angles, encouraging the musculoskeletal model to align its arm posture with the reference motion. The reward is computed as:

$$r_{\text{arm}} = \exp\left(-\sum_{j \in \text{arm}} \left(\|\hat{\mathbf{q}}_j - \mathbf{q}_j\|/\sigma_{\text{arm}}\right)^2\right) \tag{16}$$

where $\hat{\mathbf{q}}_j$ denotes the joint angle of joint j $j$ in the reference motion $M_{\text{ref}}$, $\mathbf{q}_j$ represents the current joint angle.

## B.4 TRAINING CURVES

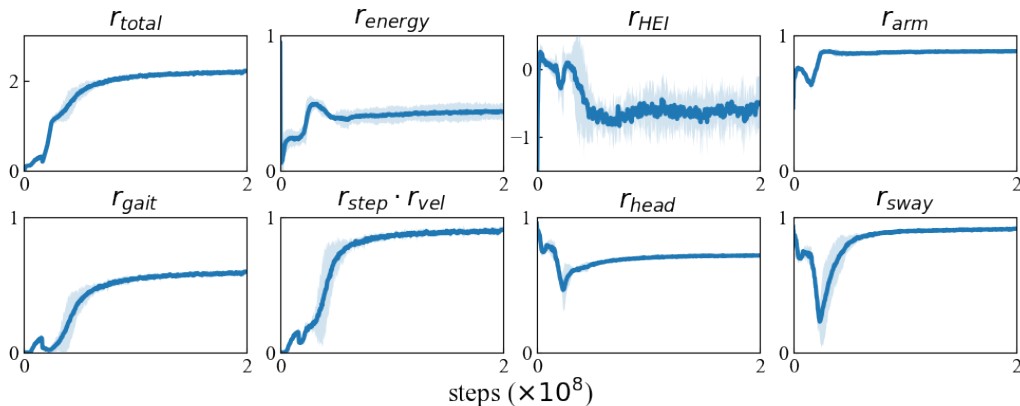

Figure B.2: Training curves of each reward component. Shaded regions indicate $\pm 2\sigma$ over $n = 5$ runs.

## C INTRAMUSCULAR REGULARIZATION ENHANCES PHYSIOLOGICAL PLAUSIBILITY OF MUSCLE ACTIVATION

In this section, we investigate how $\mathcal{L}_{\text{IMR}}$ affects both the training dynamics and the actual activations computed by the MCN. As shown in Figure C.3, during the training process with the intramuscular regularizer (IMR), the IMR-specific loss decreased steadily while the activation discrepancies across line segments of the same anatomical muscle also diminished. In contrast, training without the IMR often led to large differences between line-muscle activations where human experimental data does not show (Ong et al., 2019). Although the baseline MCN loss includes an activation regularizer that prevents over-reliance on individual muscles, this alone was insufficient to avoid highly imbalanced segment activations. Such imbalances are physiologically implausible, since humans cannot selectively control individual line segments of a single muscle. Therefore, the IMR is essential for producing activation patterns that are both stable and physiologically plausible.

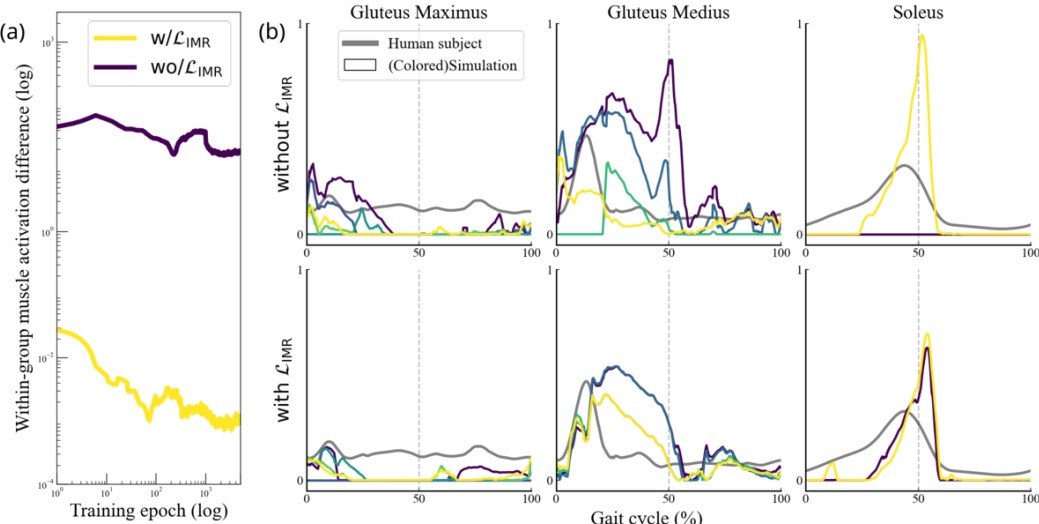

Figure C.3: $\mathcal{L}_{\text{IMR}}$ over training epochs (a), Line-wise activation profiles (colored) across the gait cycle, in comparison with human data (b)

# D   DETERMINING METABOLIC ENERGY EXPENDITURE PARAMETERS

We selected the MEE exponent parameter pair $(\alpha, \beta) = (1.5, 1.0)$. For each pair, we trained multiple data generators with different random seeds and computed the resulting PWS using Algorithm 2. Figure D.4 shows the result which revealed that $\alpha$ had a significant influence on the simulated preferred walking speed (PWS), whereas $\beta$ had relatively little effect. Therefore, we chose $\alpha = 1.5$, which produced a PWS closest to the human reference value ($v_{\text{real}} = 1.25$ m/s). For $\beta$, we selected a value of 1.0 to ensure that metabolic cost scaled linearly with muscle mass, allowing the total cost to remain consistent regardless of whether mass was considered per-muscle or globally.

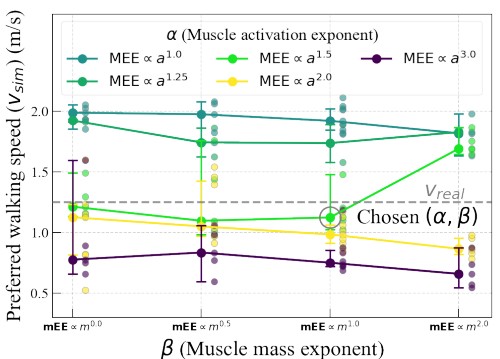

Figure D.4: Selection of MEE exponents parameter pair $(\alpha, \beta)$ based on resulting PWS.

# E   LIMITATIONS OF GAUSSIAN PROCESSES AS A SURROGATE MODEL

## E.1   GAUSSIAN PROCESSES CONSUME EXCESSIVE MEMORY

We conduct a feasibility comparison between Gaussian processes (GP) and neural network (NN) by analyzing peak runtime memory consumption. For a fair comparison, we employed a GPU-based GP implementation (Gardner et al., 2018). All experiments were conducted on an RTX 3070 GPU. Since GP require $O(N^2)$ memory, datasets larger than 20k samples exceeded the GPU VRAM limit (8GB).

| dataset size | GP (MB) | NN (MB) |
|---|---|---|
| 20,000 | Out-of-memory | 538 |
| 10,000 | 2232 | 360 |
| 500 | 258 | 226 |

Table E.1: Comparison of peak runtime memory usage between GP and NN.

## E.2   DATASET SIZE REQUIRED FOR PLAUSIBLE RESULTS EXCEEDS GP'S COMPUTATIONAL LIMIT

Figure E.5 illustrates the relationship between optimized exoskeleton parameters and the size of the training dataset when using a neural network as a surrogate. With 5k and 10k sample sizes, the resulting optimization landscapes are noisy and contain unrealistic local minima, making them unsuitable for robust parameter selection. Increasing the dataset size to 20k samples improves the smoothness and plausibility of the parameter predictions; however, the memory requirements quickly become prohibitive for GP (see Appendix E.1). In contrast, neural network surrogates can efficiently utilize large datasets with relatively low computational resources, enabling stable and physiologically plausible optimization results.

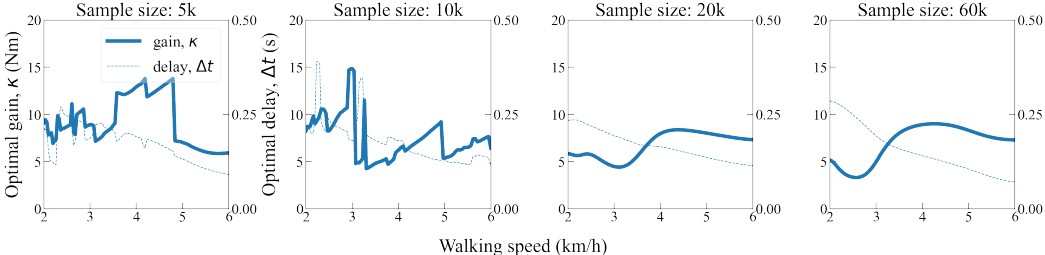

Figure E.5: Exoskeleton controller optimization results across different dataset sizes (5k, 10k, 20k, and 60k parameter samples).

## F    HUBER LOSS AND LATIN HYPERCUBE SAMPLING ARE REQUIRED FOR SMOOTH SURROGATE NETWORK LANDSCAPE

Figure F.6 shows the surrogate network landscape across walking speeds. Our proposed combination of Huber loss and Latin Hypercube Sampling (LHS) produces the smoothest landscape compared to other approaches. The MSE-based surrogate network exhibited excessive sensitivity to noisy gait data, resulting in jagged cost landscapes, especially at higher exoskeleton gains ($\kappa$), where external assistance amplifies gait variability. Similarly, the surrogate network trained on uniformly sampled data displayed aliasing artifacts due to poor sample coverage, despite using the same sample size ($N = 60k$).

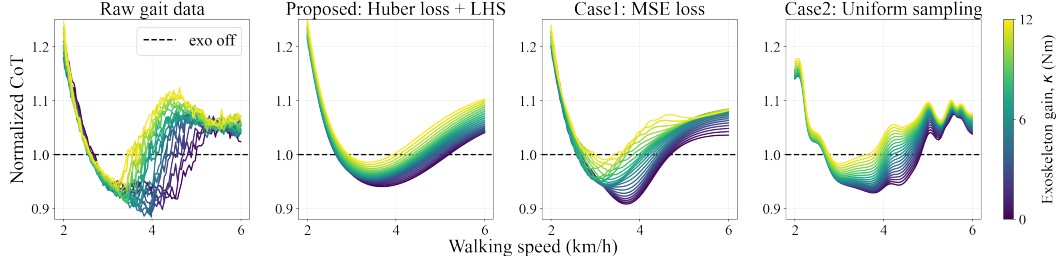

Figure F.6: Landscape of CoT from gait data generator across walking speeds and exoskeleton gains ($\kappa$) normalized to the exoskeleton-off condition (first panel). Surrogate predictions under different training setups: Our proposed configuration (second panel) yields a smoother landscape than alternative approaches (third and forth panels).

## G    MOTIVATION OF THE GRADIENT PENALTY LOSS OF THE EXOSKELETON OPTIMIZER

This penalty is motivated by the Lipschitz condition:

$$
\begin{aligned}
\|f(x_1) - f(x_2)\| &\leq \|x_1 - x_2\| \qquad \forall x_1, x_2 \in \mathcal{X}, \\
\frac{\|f(x_1) - f(x_2)\|}{\|x_1 - x_2\|} &\leq 1, \\
\lim_{x_2 \to x_1} \frac{\|f(x_1) - f(x_2)\|}{\|x_1 - x_2\|} &= \|\nabla f(x_1)\| \leq 1.
\end{aligned}
\tag{17}
$$

A small gradient norm implies local Lipschitz continuity, and a uniform bound on the gradient norm ensures global Lipschitz continuity (Gulrajani et al., 2017). Enforcing this property prevents abrupt shifts in predicted CoT values across the parameter space and yields stable gradients that can be exploited by downstream optimization.

## H    OPTIMIZING EXOSKELETON WITH SURROGATE NETWORK

Let $\{g_n\}_{n=1}^{M}$ denote the set of gait parameters. $\kappa_n$ and $\Delta t_n$ are the optimal parameters corresponding to each $g_n$, and the optimization is performed with the following objective:

$$
\begin{aligned}
\mathcal{L}(\kappa_n, \Delta t_n) = \sum_{n=1}^{M} \hat{y}(g_n;\ \kappa_n,\ \Delta t_n) \\
+ \lambda_1 \sum_{n=1}^{M} |\kappa_n| + \lambda_2 \sum_{n=1}^{M-1} \Big( |\kappa_{n+1} - \kappa_n| + |\Delta t_{n+1} - \Delta t_n| \Big).
\end{aligned}
\tag{18}
$$

Here, $\hat{y}$ represents the surrogate-predicted metabolic cost of transport. However, surrogate predictions can exhibit minor fluctuations across the parameter space due to the inherent complexity of the energy landscape or errors introduced during model training. These fluctuations can lead to abrupt variations in the exoskeleton control parameters when optimizing solely for CoT. To miti-

gate this issue, we introduce two additional regularization terms. The first regularizer discourages extreme parameter magnitudes, while the second promotes smooth variation of solutions across speeds. Together, these regularizers ensure that the optimized parameters minimize metabolic cost while remaining realistic and consistent across varying gait parameters. All hyperparameters used in the exoskeleton optimizer are provided in Appendix T.

# I    SIMULATION MODEL

## I.1    SIMULATED CHARACTER

Key muscle parameters, including isometric force and tendon slack length, were adjusted to match the OpenSim model by Rajagopal et al. (2016), and muscle mass distributions followed the estimates of Handsfield et al. (2014). The simulated subject's body mass and height were set to match average values reported in the target exoskeleton study (Lim et al., 2019a). Anthropometric data are provided in Table I.2.

| Item | Unit | Value |
|------|------|-------|
| Height | cm | 169 |
| Mass | kg | 72.9 |

Table I.2: Physical properties of the musculoskeletal character.

## I.2    EXOSKELETON

We adopted Samsung Electronics' gait enhancing and motivating system (GEMS) (Lim et al., 2019b;a). See Table I.3 for mechanical properties.

| Item | Unit | Value |
|------|------|-------|
| Body segment | kg | 1.7 |
| Thigh segment | kg | 0.6 |
| Total mass* | kg | 2.3 |

Table I.3: Physical properties of the exoskeleton.

*The total mass differs slightly from Lim et al. (2019a) because we imposed a minimum mass of 50 g on each exoskeleton element to prevent numerical instability caused by nearly singular dynamics when solving Newton's equations.*

In the human controller of the data generator, the stable PD controller computes the final desired joint torque as:

$$\tau_{\text{MCN}}^{(j)} = \tau_{\text{PD}}^{(j)} - \tau_{\text{exo}}^{(j)} \quad \forall j \in \mathcal{J}_{\text{hip}}. \tag{19}$$

where $\tau_{\text{exo}}$ denotes the torque applied by the exoskeleton, $\tau_{\text{PD}}$ denotes computed torque from the PD-target in the PD-controller and $\tau_{\text{MCN}}$ serves as the input to the MCN for computing muscle activations. This formulation ensures that the MCN receives only the net torque required for joint motion, effectively reflecting the dynamics under exoskeletal assistance.

## J SIMULATION PARAMETERS

Parameters for the simulation are displayed in Table J.4. Gait parameters and exoskeleton control parameters are uniformly sampled, while muscle parameters are sampled in an adaptive manner (Won & Lee, 2019). Adaptive sampling is a training technique in reinforcement learning that adjusts the parameter sampling rate during RL training based on the learning difficulty of each parameter. Specifically, the muscle parameter sampling interval is divided into 8 bins. For each bin, we evaluate the average episode length of the human controller while training. If some bins have lower average episode lengths than other bins, we increase the sampling rate for those bins.

| Category | Name | Unit | Range |
|---|---|---|---|
| **Muscle parameter** | **Muscle weakness** (maximum isometric force) | | |
| | Triceps surae (Calcaneous) | N.A. | $[0, 0.4]$ |
| | Tibialis anterior (Footdrop) | N.A. | $[0, 0.5]$ |
| | Gluteal muscles (Waddling) | N.A. | $[0, 0.4]$ |
| | **Muscle contracture** (optimal muscle fiber length) | | |
| | Triceps surae (Equinus) | N.A. | $[0.73, 1]$ |
| | Psoas (Crouch) | N.A. | $[0.55, 1]$ |
| **Gait parameter** | Step length | cm | $[13.4, 93.8]$ |
| | Step frequency | spm | $[1.27, 2.55]$ |
| **Exoskeleton control parameter** | Gain $(\kappa)$ | Nm | $[0, 21]$ |
| | Delay $(\Delta t)$ | sec | $[0, 0.5]$ |

Table J.4: Simulation parameter name and its range.

# K    GENERATED PATHOLOGICAL GAIT KINEMATICS

Figure K.7: Comparison of simulation results between normal muscle conditions and each pathological condition.

Exo-plore was trained with pathological muscle conditions using parameters specified in Appendix J, and the resulting gait patterns are presented in Figure K.7. Rendered images with descriptive annotations are provided to facilitate understanding of each pathological condition. Close-up images of the feet include ground reaction force visualization to illustrate contact states during the gait cycle.

We compare pathological gaits with unimpaired gait to clearly show deviation patterns.

For pathological gait cases, the walking speed was fixed at 3.5 km/h to ensure consistent comparison across severities. In contrast, for unimpaired gait, we used the preferred walking speed (PWS) identified by Algorithm 2, and selected gait parameters that minimized the CoT at that speed.

| Name | Cause | Description |
|------|-------|-------------|
| Equinus | Contracture in Triceps surae | Walking on the toes due to persistent plantarflexion of the ankle. The heel does not touch the ground during gait. |
| Waddling | Weakness in Gluteus | Swaying of the trunk from side to side while walking due to weakness of the gluteal muscles, which fail to stabilize the pelvis. This gait is also referred to as a waddling or "duck-like" gait. |
| Crouch | Contracture in Psoas | Walking in a flexed, crouched posture, which maintains the hips in persistent flexion, often accompanied by knee flexion. |
| Calcaneous | Weakness in Triceps surae | Walking with the heel contacting the ground first due to weakness of the triceps surae muscles, leading to excessive dorsiflexion of the ankle. |
| Footdrop | Weakness in Tibialis anterior | Inability to dorsiflex the ankle during foot-off, resulting in the toes dragging on the ground. Patients often compensate with high-stepping gait. |

Table K.5: Types of pathological gait patterns, their causes, and descriptions.

## L OVERACTIVATION OF THE RECTUS FEMORIS IN THE SIMULATION

Figure 3 shows the muscle activation of rectus femoris (RF) have the largest sim-to-real gap. Simulated RF is activated much more than humans typically utilize it, especially around the foot-off phase. This increased activation of the RF in the gait data generator is related to the superior motor performance of the human controller, as mentioned earlier. Humans tend to control the RF together with the vastus lateralis and vastus medialis, as these muscles are located close to each other and work in coordination, rather than using it independently. Consequently, human activation data shows that these three muscles are activated in similar patterns. However, the RF, being a biarticular muscle, can move two joints and is more metabolically efficient when activated alone. This provides an incentive for the controller to actively utilize it. To investigate this, we trained the human controller while artificially deactivating the RF, and while kinematics remained nearly unchanged, we observed a slightly higher ($\sim 4\%$) CoT (See Figure L.8). Therefore, to address the excessive use of the RF, an additional regularizer that reflects the limitations of human motor control, such as synergy control (Tresch et al., 1999; Ting & Chvatal, 2010; Banks et al., 2017; Shourijeh & Fregly, 2020), should be applied.

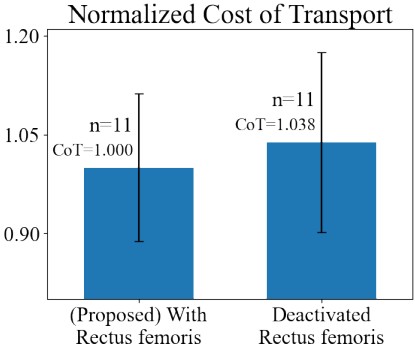

Figure L.8: Inclusion of rectus femoris in the model reduces normalized cost of transport compared to the deactivated rectus femoris condition.

## M GROUND REACTION FORCE OF UNASSISTED GAIT

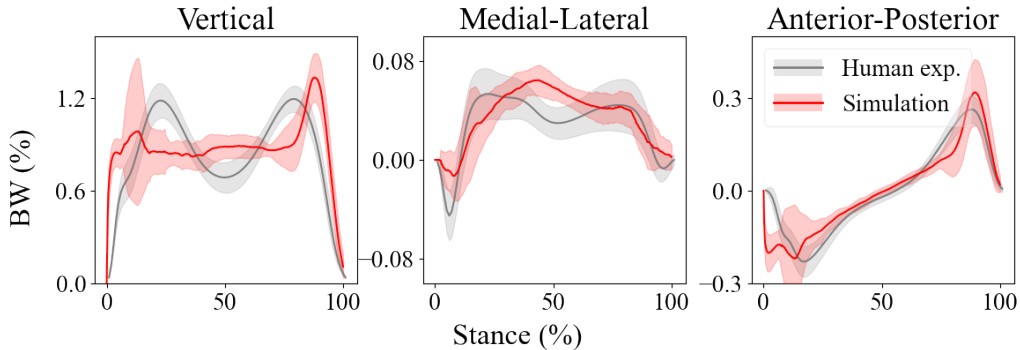

Figure M.9: Comparison of simulated and experimental ground reaction forces (GRF).

Figure M.9 compares the simulated and experimental ground reaction forces (GRF) (Horst et al., 2021) over a single stance cycle, with all forces normalized by body weight (BW) for consistency with the reference human data. Overall, the simulated GRF profiles exhibit characteristic patterns similar to those observed in human locomotion. The primary discrepancy appears in the vertical component, where the simulation produces a sharper peak than the experimental measurements. This difference likely arises from structural modeling: while the human foot features an arched and compliant architecture that distributes load more gradually, our simulation employs a box-shaped rigid element, resulting in a more abrupt transfer of force during stance.

## N  ALTERNATIVE HEI REWARDS

- **Assist Maximization:** Encourages the agent to align with assistive power by promoting positive power contributions:

$$r_{\text{HEI}} = r_{\text{assist}} = \frac{1}{Z} \sum_{s \in \{L,R\}} \max(0, P_s) \tag{20}$$

- **No HEI Reward:** With $r_{\text{HEI}} = 0$, the human is modeled as minimizing metabolic energy expenditure without incorporating any additional considerations related to exoskeletal assistance.

## O   HEI Makes Higher Metabolic Reduction Rate than the Alternative Models

Figure 7 demonstrates that our HEI reward accurately predicts the metabolic reduction rate (MRR) achieved through exoskeleton assistance. This finding may appear counterintuitive, as the HEI reward does not explicitly regulate metabolic energy expenditure. Furthermore, one might expect the HEI reward to impede metabolic reduction by exoskeleton, given its potential competition with $r_{energy}$, which directly minimizes metabolic energy consumption. To investigate this apparent paradox, we analyze the relationship between gait parameters and their metabolic benefits as follows:

$$\text{Benefit} = \text{CoT}(\kappa = 0) - \min\big(\text{CoT}(\kappa > 0)\big) \tag{21}$$

Figure O.10 reveals that the maximum normalized Benefit remains consistent across different HEI reward configurations. While the mean values are nearly identical across reward conditions, the gait parameters at which these maxima occur differ significantly. In the absence of HEI, the maximum Benefit coincides with the highest cost of transport (CoT) values. This observation aligns with expectations, as $r_{energy}$ primarily aims to reduce MEE, leading the human controller to learn exoskeleton utilization strategies at parameters corresponding to maximum CoT.

In contrast, when our proposed HEI reward is employed, the maximum benefit occurs at large gait parameters (Step frequency=2.5 Hz, Step length=0.6 m, walking speed=5.2 km/h) than former. This phenomenon can be attributed to the enhanced effectiveness of the HEI reward at higher walking speeds, where exoskeleton assistance power is relatively elevated. Consequently, the human controller learns to exploit exoskeleton assistance more effectively under these conditions, resulting in higher measured MRR values despite the lower baseline CoT. This finding is consistent with human experimental studies (Lim et al., 2019b), which report maximum MRR at relatively high walking speeds of 4 km/h.

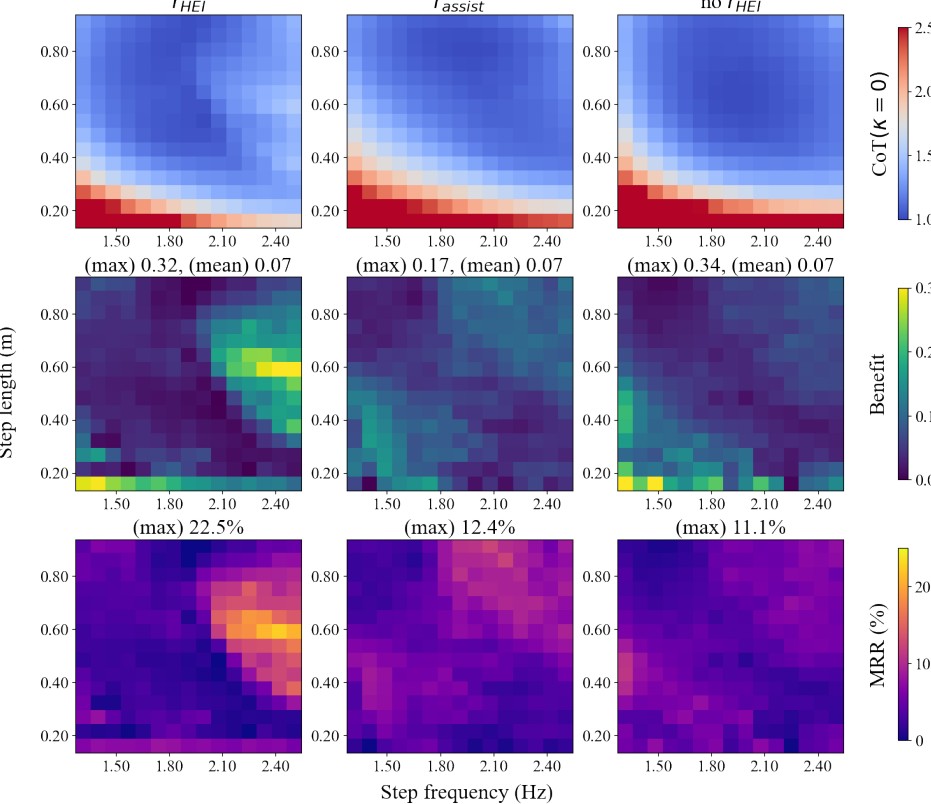

Figure O.10: Comparison of the simulated cost of transport (top), Benefit (middle) and metabolic reduction rate (bottom) between our proposed HEI reward($r_{HEI}$) and the alternatives ($r_{assist}$, no $r_{HEI}$).

## P  OPTIMAL EXOSKELETON GAIN TRENDS IN PATHOLOGICAL GAIT

### P.1  PATHOLOGY-SPECIFIC TRENDS IN OPTIMAL GAIN

Although the absolute scale of the optimal gain differed between pathological gait data generator, the gains consistently follow severity-dependent trends. This suggests that pathology-specific gain patterns can serve as distinguishing characteristics, since the gain trend is less sensitive to surrogate hyperparameters than the absolute gain value. Based on these trends, we identified two distinct groups reflecting fundamentally different assistance requirements:

**Group A (Benefit from hip exoskeleton).**  Crouch and calcaneus gaits show increasing optimal gains with severity. Crouch gait, caused by psoas contracture, maintains stable but energetically expensive postures with sustained knee and hip flexion. Similarly, calcaneus gait from triceps surae weakness results in heel-walking, relying heavily on hip flexion to compensate for the inability to plantarflex the ankle. Both conditions feature stable but metabolically demanding gait patterns where exoskeleton assistance effectively reduces energy expenditure without compromising stability.

**Group B (Limited benefit from hip exoskeleton).**  Equinus and waddling gaits exhibit decreasing optimal gains with severity. Equinus, resulting from triceps surae contracture, causes inherently unstable toe-walking where external forces exacerbate balance issues. Waddling gait, stemming from gluteus weakness, involves lateral trunk sway for pelvic stabilization, where compensatory strategy that external hip assistance can disrupt. These pathologies prioritize maintaining precarious stability over energy optimization, making them less amenable to hip exoskeleton intervention.

Footdrop presents a unique case with no clear severity correlation. While hip exoskeleton assistance was expected to benefit the compensatory high-stepping pattern, the frequent toe-ground collisions during toe-off introduced substantial gait variability, preventing stable optimization convergence. Detailed discussion is provided in Appendix P.2.

This classification reveals a fundamental principle: pathologies with stable but energy-intensive compensation patterns (Group A) benefit from increasing assistance, while those with unstable compensation mechanisms (Group B) require minimal intervention to preserve their delicate balance strategies. Thus, effective exoskeleton prescription depends not on pathology severity alone, but on the underlying stability-energy trade-off inherent to each condition's biomechanical adaptation.

### P.2  THE CAUSE OF UNCLEAR TREND OF FOOT DROP IS EXCESSIVE GAIT VARIABILITY OF FOOT DROP

Foot drop did not follow a clear trend due to its inherently unstable gait dynamics. This pathology, characterized by insufficient toe clearance from tibialis anterior weakness, leads to frequent toe-ground collisions that substantially increase gait variability. As shown in Figure P.11, severe foot drop exhibited a coefficient of variation (CV) in CoT nearly four times higher than that of other pathologies. While most conditions maintained consistent CV values across severity levels, foot drop's variability increased dramatically with severity. This elevated stochasticity in the simulated gait, arising from unpredictable toe-catching events, obscures any systematic relationship between pathology severity and optimal exoskeleton gain.

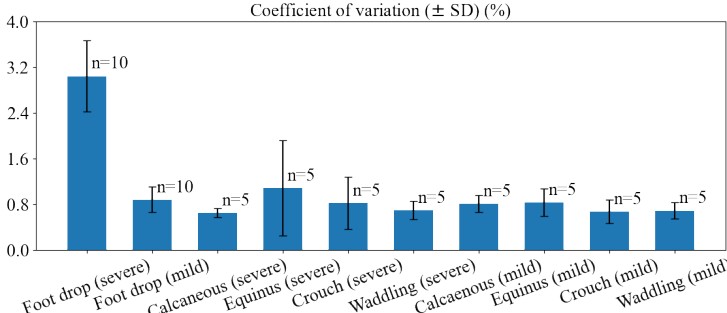

Figure P.11: The foot drop pathology exhibits the highest coefficient of variation among all pathology types.

P.3   EFFECT OF HEI REWARD ON OPTIMAL GAIN TRENDS

The HEI reward ($r_{\text{HEI}}$) proved essential for capturing clinically relevant exoskeleton optimization outcomes. For example, calcaneus gait, which should benefit from assistance due to its high hip muscle demands, only showed the expected positive correlation (Group A) when $r_{\text{HEI}}$ was included. Without this reward term, calcaneus incorrectly appeared in Group B with decreasing gains, despite maintaining strong linearity. This demonstrates that $r_{\text{HEI}}$ is a necessary component for revealing the true biomechanical characteristics of pathological gait, ensuring that optimization captures the actual assistance needs.

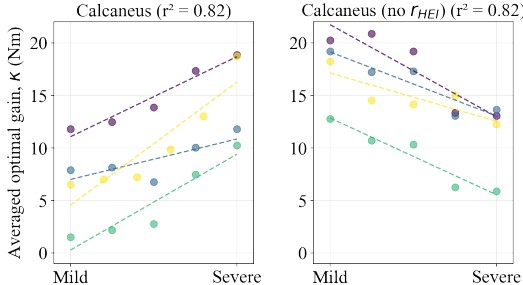

Figure P.12: Our proposed HEI reward reverses the average optimal gain trend with respect to pathology severity.

P.4   $R^2$ VALUES FOR LINEAR FITS OF OPTIMAL GAIN ACROSS PATHOLOGY SEVERITIES

In the Table P.6, we examined the linear relationship between optimal exoskeleton gain and pathology severity by calculating Pearson's R² values across different random seeds.

| Category | Seed | $R^2$ | Category | Seed | $R^2$ |
|---|---|---|---|---|---|
| **Equinus** | seed1 | 0.972 | **Calcaneus** | seed1 | 0.944 |
| | seed2 | 0.914 | | seed2 | 0.589 |
| | seed3 | 0.787 | | seed3 | 0.885 |
| | seed4 | 0.941 | | seed4 | 0.845 |
| **Waddling** | seed1 | 0.574 | **Foot drop** | seed1 | 0.509 |
| | seed2 | 0.930 | | seed2 | 0.711 |
| | seed3 | 0.991 | | seed3 | 0.808 |
| | seed4 | 0.825 | | seed4 | 0.321 |
| **Crouch** | seed1 | 0.949 | **Calcaneus** | seed1 | 0.819 |
| | seed2 | 0.611 | (no $r_{HEI}$) | seed2 | 0.845 |
| | seed3 | 0.875 | | seed3 | 0.923 |
| | seed4 | 0.956 | | seed4 | 0.708 |

Table P.6: Pearson's R² values for linearity between optimal exoskeleton gain and pathology severity.

## Q  ANTHROPOMORPHIC CONSISTENCY BETWEEN REAL AND SIMULATED DATA

To validate the consistency between simulation and experiment, we confirmed that the anthropometric parameters of our simulated character match those of subjects in the real datasets. As shown in Table Q.7, the model's height and weight fall within the natural variability of the populations in all referenced datasets, indicating that these datasets are appropriate for use in our comparisons.

| Dataset | Height (cm) | Weight (kg) | n= |
|---|---|---|---|
| **Simulation** | 169 | 72.9 | – |
| Kinematics (Boo et al., 2025) | $173.2 \pm 5.6$ | $74.1 \pm 12.5$ | 120 |
| GEMS (Lim et al., 2019b) | $173 \pm 8.2$ | $68.4 \pm 6.3$ | 5 |
| GRF (Horst et al., 2021) | $176 \pm 9$ | $70.7 \pm 12.0$ | 350 |

Table Q.7: Comparison between the anthropometric parameters of the simulated character and real human datasets.

## R  LIMITATIONS OF HILL TYPE MUSCLE DYNAMICS.

The Hill muscle model, which was originally proposed based on experiments on frog muscle in 1938 (Hill, 1938), has since been used as the mathematical description of muscle in almost all musculoskeletal simulation studies. In this section, we discuss the limitations of the simulation that arise from the known shortcomings of the Hill muscle model.

The contractile force of a Hill-type muscle is formulated as follows:

$$f = f(l, \dot{l}, a) = a \cdot f_l(l) \cdot f_v(\dot{l}) + f_p(l),$$

where $f_l$, $f_v$, and $f_p$ represent the force–length, force–velocity, and passive force–length relationships, respectively. The variables $l$ and $a$ denote the muscle fibre length and activation level. Hill's original experiments imply that these three factors—activation, length, and shortening velocity—contribute independently to muscle force generation. Despite its wide adoption, several important limitations of the Hill model have been identified.

**Inability to reproduce history-dependent behavior.** Real muscles exhibit clear history-dependent behavior. When a muscle is stretched, the resulting contractile force exceeds the force produced when the muscle is activated directly at the same length. This phenomenon is known as force enhancement following stretch. Conversely, active shortening results in force depression. Because the Hill muscle model is stateless, it cannot reproduce these history-dependent properties (Abbott & Aubert, 1952; Nishikawa et al., 2012).

This limitation becomes particularly problematic in the simulation of fast running. During high-speed locomotion, muscles such as the hamstrings and calf muscles are activated while being stretched at ground contact. In real muscles, this stretch-induced force enhancement allows them to sustain substantially higher forces at a lower energetic cost, enabling rapid and efficient running. In contrast, Hill-based simulations underestimate the available force and overestimate metabolic expenditure, thereby producing a significant sim-to-real gap.

**Lack of activation-dependent changes in muscle parameters.** Experimental studies have shown that key muscle parameters, such as optimal fibre length (Roszek & Huijing, 1997) and maximum shortening velocity (Chow & Darling, 1999), depend on the activation level of the muscle. However, in the Hill muscle model these quantities are typically treated as fixed constants.

## S QUANTITATIVE COMPARISON BETWEEN SIMULATED AND EXPERIMENTAL DATA

To evaluate the similarity between the simulated and experimental time-series data, we report three metrics: the normalized root-mean-square error (NRMSE), the Pearson correlation coefficient ($R$), and a normalized dynamic time warping (NDTW) distance.

The NRMSE quantifies the overall magnitude error between the two signals and is normalized by the range of the experimental data, allowing comparisons across different variables. Lower NRMSE values indicate better amplitude agreement. The $R$ measures the similarity in waveform shape between the signals, independent of scale. Values closer to 1 indicate stronger similarity in the temporal pattern. The NDTW distance assesses temporal alignment between the two signals while allowing local time stretching or compression. The distance is normalized by the amplitude range and warping-path length, so smaller values indicate closer temporal correspondence.

### S.1 UNASSISTED GAIT

| Joint | NRMSE | $R$ | NDTW |
|---|---|---|---|
| Ankle Dorsiflexion | 0.563 | 0.892 | 0.200 |
| Knee Flexion | 0.185 | 0.966 | 0.108 |
| Hip Flexion | 0.210 | 0.965 | 0.068 |
| Hip Abduction | 0.244 | 0.777 | 0.067 |
| Hip Internal Rotation | 1.436 | -0.443 | 1.019 |

Table S.8: Similarity metrics between simulated and real joint kinematics.

| Muscle | NRMSE | $R$ | NDTW |
|---|---|---|---|
| Bicep Femoris Longus | 0.799 | 0.835 | 0.266 |
| Bicep Femoris Short | 0.533 | -0.216 | 0.344 |
| Gastrocnemius Lateral Head | 0.531 | 0.775 | 0.129 |
| Gastrocnemius Medial Head | 0.348 | 0.605 | 0.057 |
| Soleus | 0.419 | 0.575 | 0.086 |
| Gluteus Maximus | 1.296 | 0.207 | 1.254 |
| Gluteus Medius | 0.283 | 0.574 | 0.025 |
| Vastus Lateralis | 0.271 | 0.634 | 0.061 |
| Vastus Medialis | 0.366 | 0.649 | 0.151 |
| Rectus Femoris | 5.565 | -0.430 | 2.040 |
| Tibialis Anterior | 1.780 | 0.516 | 0.395 |
| Peroneus Brevis | 0.321 | 0.823 | 0.097 |
| Peroneus Longus | 0.223 | 0.931 | 0.072 |
| Semitendinosus | 0.596 | 0.353 | 0.117 |

Table S.9: Similarity metrics between simulated and real muscle activations.

| GRF Axis | NRMSE | $R$ | NDTW |
|---|---|---|---|
| Vertical | 0.216 | 0.563 | 0.057 |
| Medial-Lateral | 0.159 | 0.807 | 0.039 |
| Anterior-Posterior | 0.113 | 0.923 | 0.028 |

Table S.10: Similarity metrics between simulated and real ground reaction forces.

## S.2 ASSISTED GAIT

| Metric | Walking velocity (km/h) | Hip flexion angle | Hip flexion velocity | Exo assist moment | Exo assist power |
|---|---|---|---|---|---|
| NRMSE | 2.0 | 0.267 | 0.247 | 0.210 | 0.294 |
|  | 3.0 | 0.241 | 0.204 | 0.158 | 0.290 |
|  | 4.0 | 0.227 | 0.187 | 0.154 | 0.311 |
|  | 5.0 | 0.242 | 0.173 | 0.111 | 0.289 |
| $R$ | 2.0 | 0.896 | 0.645 | 0.916 | 0.328 |
|  | 3.0 | 0.922 | 0.775 | 0.962 | 0.754 |
|  | 4.0 | 0.887 | 0.856 | 0.932 | 0.507 |
|  | 5.0 | 0.867 | 0.927 | 0.969 | 0.737 |
| NDTW | 2.0 | 0.120 | 0.067 | 0.089 | 0.113 |
|  | 3.0 | 0.106 | 0.068 | 0.058 | 0.105 |
|  | 4.0 | 0.074 | 0.072 | 0.051 | 0.129 |
|  | 5.0 | 0.066 | 0.078 | 0.033 | 0.102 |

Table S.11: Similarity metrics between simulated and real kinematics and dynamics.

| Data Type | $r_{\text{HEI}}$ | $r_{\text{assist}}$ | no $r_{\text{HEI}}$ |
|---|---|---|---|
| RMS Assist Moment | 0.964 | 0.745 | 0.726 |
| Mean Assist Power | 0.462 | -0.180 | -0.082 |
| Mean Resist Power | 0.919 | 0.623 | 0.589 |

Table S.12: Correlation (r) between simulated and real human–exoskeleton interaction.

## T HYPERPARAMETERS

### T.1 HUMAN CONTROLLER

| **Reward parameters** | | | | | | | |
|---|---|---|---|---|---|---|---|
| $w_{\text{gait}}$ | 1.0 | $w_{\text{energy}}$ | 0.35 | $w_{\text{arm}}$ | 1.0 | $w_{\text{HEI}}$ | 0.1 |
| $\sigma_{\text{step}}$ | 1.581 | $\sigma_{\text{vel}}$ | 3.162 | $\sigma_{\text{head}}$ | 4.0 | $\sigma_{\text{sway}}$ | 0.816 |
| $\sigma_{\text{arm}}$ | 1.0 | $\lambda_r$ | 0.018 | $\lambda_v$ | 0.0105 | $\lambda_\omega$ | 0.045 |
| $\delta_{\text{pelvis}}$ | 10° | $\delta_{\text{spine}}$ | 3° | $\delta_{\text{foot}}$ | 12° | $k_{\text{energy}}$ | 0.2 |
| $k_{alive}$ | 0.1 | | | | | | |
| **PPO parameters for PDN** | | | | | | | |
| Batch size | 8192 | $lr$ | 5e-5 | $\log \sigma$ | 1.0 | $\gamma$ | 0.99 |
| $\lambda$ | 0.99 | $e_{\text{clip}}$ | 0.2 | $kl_{\text{coeff}}$ | 0.01 | $kl_{\text{target}}$ | 0.01 |
| **MCN learning parameters** | | | | | | | |
| $N_{\text{epoch}}$ | 5 | $lr$ | 1e-4 | $w_{\text{reg}}$ | 0.01 | $w_{\text{IMR}}$ | 0.1 |

### T.2 EXOSKELETON OPTIMIZER

| $\delta_h$ | 1.0 | $\lambda_{\text{grad}}$ | 5e-2 | $\lambda_{\text{L1}}$ | 5e-4 | $\lambda_{\text{L2}}$ | 5e-4 |
|---|---|---|---|---|---|---|---|
| Epoch | 1e4 | Hidden | [256, 256] | $lr_{\text{init}}$ | 0.1 | $lr_{\text{end}}$ | 5e-4 |

# U   Nomenclature

## U.1   Human controller and exoskeleton

| | |
|---|---|
| $\tau_{\text{exo}}$ | Hip assistive torque generated by the exoskeleton controller. |
| $\kappa$ | Gain scaling the magnitude of exoskeleton torque. |
| $\Delta t$ | Time delay of the delayed-output feedback controller. |
| $u(t)$ | Control signal encoding relative hip motion. |
| $\theta_r, \theta_l$ | Low-pass filtered right and left hip joint angles. |
| $s$ | Full simulation state. |
| $s_{\text{skeleton}}$ | State of the musculoskeletal skeleton (kinematics and dynamics). |
| $s_{\text{muscle}}$ | Muscle-related state (forces, bounds, and dynamic parameters). |
| $s_{\text{exo}}$ | Exoskeleton state, including recent assistive torques. |
| $q_d$ | PD target joint positions produced by PoseNet. |
| $a$ | Muscle activations applied to the musculoskeletal model. |
| $\tau(a)$ | Joint torque resulting from muscle activations as computed by the simulation model. |
| $\tau_{\text{PD}}$ | Output torque of the PD controller (before subtracting exoskeleton torque). |
| $\tau_{\text{MCN}}$ | Target joint torque fed into the MCN to compute muscle activations. |

## U.2   RL Reward

| | |
|---|---|
| $r_{\text{total}}$ | Total reinforcement learning reward for the PoseNet. |
| $r_{\text{gait}}$ | Gait tracking reward composed of step, velocity, head, and sway terms. |
| $r_{\text{step}}$ | Reward encouraging target step length. |
| $r_{\text{vel}}$ | Reward encouraging target walking speed. |
| $r_{\text{head}}$ | Reward encouraging stable head orientation and motion. |
| $r_{\text{sway}}$ | Reward penalizing abnormal body sway. |
| $r_{\text{arm}}$ | Upper-arm imitation reward encouraging natural arm posture. |
| $r_{\text{energy}}$ | Reward regularizing metabolic energy expenditure. |
| $r_{\text{HEI}}$ | Human–exoskeleton interaction reward based on resistance minimization. |
| $h$ | Current foot position. |
| $h_{\text{desired}}$ | Target foot position corresponding to the desired step length. |
| $v$ | Average forward walking velocity of the character. |
| $v_{\text{desired}}$ | Target forward walking velocity. |
| $\sigma_{\text{step, vel}}$ | Scale parameter for $r_{\text{step, vel}}$ |
| $\theta_{\text{head}}$ | Head orientation with respect to the progression and global up directions. |
| $a_{\text{lin}}$ | Linear acceleration of the head. |
| $a_{\text{ang}}$ | Angular acceleration of the head. |
| $k_{\text{alive}}$ | Baseline term preventing full suppression of the head reward. |
| $\lambda_r, \lambda_v, \lambda_\omega$ | Thresholds for head angle, linear acceleration, and angular acceleration. |
| $\sigma_{\text{head}}$ | Scale parameter used in the head-stability reward. |
| $q_b$ | Orientation angle of body segment $b$. |
| $\bar{q}_b$ | Reference body angle from normative gait data. |
| $\delta_b$ | Allowed deviation threshold for segment $b$. |
| $B$ | Set of body segments considered in the sway reward. |
| $r_{\text{sway},b}$ | Sway reward term for body segment $b$. |
| $\hat{q}_j$ | Reference joint angle of upper-arm joint $j$ in the reference motion. |
| $q_j$ | Current joint angle of upper-arm joint $j$. |
| $\sigma_{\text{arm}}$ | Scale parameter for arm imitation error. |
| $P_k$ | Mechanical power applied by the exoskeleton on side $k \in \{L, R\}$. |

## U.3   RL action

| | |
|---|---|
| $\Delta\phi$ | Phase increment controlling advancement or delay of the gait phase. |
| $\Delta M$ | Residual pose displacement applied to the reference pose. |
| $\phi_t$ | Gait phase at time step $t$. |
| $M_t$ | Target pose for the PD controller at time step $t$. |
| $M_{\text{ref}}$ | Reference gait motion from which residuals are applied. |
| $\oplus$ | Operator combining a reference pose with a residual displacement. |

## U.4 MCN LOSS AND INTRAMUSCULAR REGULARIZATION

| | |
|---|---|
| $L_{\text{MCN}}$ | Total loss for training the muscle coordination network. |
| $L_{\text{IMR}}$ | Intramuscular regularization loss enforcing coherent line-muscle activations. |
| $w_{\text{reg}}$ | Weight for activation regularization in the MCN loss. |
| $w_{\text{IMR}}$ | Weight for the intramuscular regularizer term. |
| $N_g$ | Number of anatomical muscle groups. |
| $G_g$ | Index set of line muscles belonging to group $g$. |
| $\bar{a}_g$ | Mean activation over all line muscles in group $g$. |

## U.5 METABOLIC ENERGY

| | |
|---|---|
| MEE | Metabolic energy expenditure. |
| $m_i$ / $a_i$ | Mass / Activation associated with the $i$-th muscle. |
| $\alpha$, $\beta$ | Exponent parameters governing effort- and fatigue-like behavior in the MEE model. |
| CoT | Metabolic cost of transport (energy per unit distance). |
| $L$ | Step length in the PWS evaluation routine. |
| $f$ | Step frequency in the PWS evaluation routine. |
| $\tau_{\text{cyc}}$ | Time interval corresponding to one gait cycle. |
| Benefit | Improvement in CoT due to assistance |

## U.6 EXOSKELETON OPTIMIZER

| | |
|---|---|
| $X$ | Simulation parameter space. |
| $x_i$ | Sampled parameter point from $X$. |
| $\mathcal{B}$ | Buffer of parameter–CoT pairs collected from simulation. |
| $c$ | Exoskeleton control parameter vector. |
| $c^*$ | Optimal exoskeleton control parameter identified by the optimizer. |
| $\hat{f}_\theta(c)$ | Surrogate network predicting CoT from control parameters $c$. |
| $L_{\text{surrogate}}$ | Training loss for the surrogate network. |
| $\hat{y}$ | Surrogate prediction of CoT. |
| $y$ | Ground-truth CoT obtained from simulation. |
| $L_{\delta_h}$ | Huber loss. |
| $\nabla_x \hat{y}$ | Gradient of surrogate output with respect to its input parameters. |
| $\lambda_{\text{grad}}$ | Coefficient for the gradient penalty term. |
| $\lambda_{L1}$, $\lambda_{L2}$ | Coefficients for $L_1$ and $L_2$ regularization on network weights. |
| $w_i$ | Individual weight parameter of the surrogate network. |
| $g_n$ | Discrete gait parameter (e.g., walking speed) index $n$. |
| $\kappa_n$ | Optimal exoskeleton gain corresponding to gait parameter $g_n$. |
| $\Delta t_n$ | Optimal time delay corresponding to gait parameter $g_n$. |
| $\lambda_1, \lambda_2$ | Regularization weights in the final exoskeleton optimization objective. |
| $M$ | Number of discrete gait parameter settings used in the optimization. |

## U.7 OTHER HYPERPARAMETERS

| | |
|---|---|
| Batch size | Number of samples per PPO update batch. |

| $lr$ | Learning rate (PPO or MCN, depending on context). |
|---|---|
| $\log \sigma$ | Log standard deviation of the action distribution. |
| $\gamma$ | Discount factor in PPO. |
| $\lambda$ | GAE parameter for advantage estimation. |
| $e_{\text{clip}}$ | PPO clipping parameter. |
| $kl_{\text{coeff}}$ | KL penalty coefficient in PPO. |
| $kl_{\text{target}}$ | Target KL divergence in PPO. |
| $N_{\text{epoch}}$ | Number of epochs for MCN training. |
| $w_{\text{IMR}}$ | Weight of the IMR term in the MCN loss. |
| Epoch | Number of training epochs for the surrogate network. |
| Hidden | Hidden-layer dimensions of the surrogate network MLP. |
| $lr_{\text{init}}/lr_{\text{end}}$ | Initial / Final learning rate of the optimizer schedule. |

