# OpenReview forum: "Exo-Plore: Exploring Exoskeleton Control Space through Human-aligned Simulation"
_ICLR.cc/2026/Conference — ICLR 2026 Poster_

### Official Review · Reviewer_BgQH · 2025-10-28

**Soundness:** 3
**Presentation:** 2
**Contribution:** 2
**Rating:** 2
**Confidence:** 4

**Summary:**

This paper proposes Exo-plore, a simulation pipeline to optimize exoskeleton controller parameters without real human experiments. In Exo-plore, the authors train a gait generator to generate gait trajectories with exoskeleton assistance. Then they train a surrogate model with the generated gait data under a set of controller parameters sampled by Latin hypercube. The final control parameter is selected by optimizing over the surrogate model. The experimental results demonstrate the qualitative fidelity of the gait generator in both joint and muscle levels compared with real human experiments, and the optimization results in both healthy and disable conditions.

**Strengths:**

1. This paper aims at conducting exoskeleton control optimization in pure simulation with biomechanical fidelity by using a musculoskeletal-based controller, which is well motivated and will benefit safer and more efficient assisted control optimization.

2. The experimental results is comprehensive with video demonstration, which demonstrate the qualitative fidelity of the gait generator and the optimized controller parameters. The limitations are also mentioned.

**Weaknesses:**

1. From the text description and Figure 1, the whole optimization pipeline seems not a closed-loop. During the training of the gait generator, the model is fit to a certain control parameter by maximizing $r_{HEI}$. This control parameter is not optimized. When training the surrogate model, different control parameters are fed to the gait generator to get gait data. I think the data collection pipeline is unreasonable to me, because control parameters sampled by Latin hypercube are not in the training distribution of gait generator, which may lead to lower $R_{HEI}$ and unknown gait generating performance.

2. Algorithm 1 seems very time-consuming, which requires whole training round of gait generator for each reward parameters.

3. The reward and loss terms in e.q. (2), (3), (4), (7) have many hyperparameters. It may be hard to determine these hyperparameters without real human experimental data. It may not be practical to apply the proposed method in pure simulation.

4. According to Figure 8 (b)-(d), the trained surrogate model seems sensitive to the weight of gradient penalty loss, where the optimized control parameters have distinct discrepancy. The "smooth landscape" shown by the fitted surrogate model might also be accessed by Gaussian process with much fewer data under proper kernel function.

5. (Minor) The writing can be further improved to be clearer. e,g. (1) In Algorithm 1 and 3, There is no formal definition of "trainDataGenerator" and "EvalCoT" in the main text. (2) In line 390: there is no formal introduction of $r_{assist}$.

**Questions:**

1. In Algorithm 1, how many candidates $(\alpha, \beta)$ are evaluated during experiment?  What is the training time of a single round of gaite generator training?

2. In Algorithm 2, how many available $(L, f)$ are evaluated?

3.  How to set the hyperparameters of the defined reward and loss functions in practice?

4. Do the body parameters of the used musculoskeletal model match the human with real experimental data (e.g. height and weight)?

5. The whole optimization problem is a 2d black box optimization problem, where GP-based Bayesian optimization can usually converge within 100 samples even under noisy function evaluation. Why did you generate over 60k data to train a surrogate model in Appendix D.2? I think this amount of data is sufficient to delineate a 2d function landscape without any further surrogate fitting.

---

> ### Author Response · Authors · 2025-11-15
>
> Thank you for your comments and helpful suggestions. We appreciate the feedback on the revisions. Please let us know if any points require further clarification or additional information.
>
> &nbsp;
>
> ---
>
> &nbsp;
>
> **W1.** The gait generator is not trained with a single fixed control parameter; instead, it is trained across the full range of control parameters used in the later optimization stage. Therefore, the surrogate-training samples are not out-of-distribution. We will revise the manuscript to clarify this point.
>
> &nbsp;
>
> ---
>
> &nbsp;
>
> **W2.** We agree that Algorithm 1 is computationally expensive due to its brute-force search. However, it can still be executed on a modest computing cluster, and its cost remains far lower than that of training large AI models such as LLMs, or VLs. Reducing this computational cost is an important direction for future work.
>
> &nbsp;
>
> ---
>
> &nbsp;
>
> **W3 + Q3.** Generating physiologically plausible gait without reference motion is inherently underconstrained, especially in exoskeleton scenarios where no reference trajectories exist (line 104). Additional shaping terms are therefore essential for achieving stable and realistic motion across diverse human conditions. The number of hyperparameters is comparable to other musculoskeletal simulation frameworks.
>
> Moreover, most hyperparameters have clear justification. Equation (3) is systematically tuned using Algorithm 1.
> - Equation (4) is determined by the requirement that the MCN produce physically consistent muscle activations, following Generative GaitNet [1].
> - Equation (7) uses standard regression-loss principles and is minimally tuned, with its influence shown in Fig. 8.
>
> &nbsp;
>
> ---
>
> &nbsp;
>
> **W4.** In Fig. 8, the landscapes are clearly distinguishable because the gradient-penalty weights differ by an order of magnitude (0.1 vs. 0.01). As discussed in the paper, this reflects an inherent tradeoff: larger weights produce more general and predictable behavior, while smaller weights allow the model to follow simulation data more precisely. This balance serves as a design parameter within our Exo-plore framework.
>
> Regarding Gaussian Processes (GPs), please refer to the answer for Q5 below.
>
> &nbsp;
>
> ---
>
> &nbsp;
>
>
> **W5.** We appreciate the reviewer’s feedback. We will revise the manuscript and formally define the missing components
>
> &nbsp;
>
> ---
>
> &nbsp;
>
>
> **Q1.** We evaluate 20 candidate parameter sets. As shown in Fig. C.2, each parameter is tested with ~5 repetitions for statistical reliability. Training one gait generator requires ~100M samples and takes ~8 hours on 128 cpu cores, with all experiments completed within one week.
>
> &nbsp;
>
> ---
>
> &nbsp;
>
> **Q2.** We discretize the parameter range in Table I.4 into 22 intervals per dimension, yielding 484 evaluations. Using 14 intervals (196 samples) gives similar optima; we selected 22 for conservative coverage.
>
> &nbsp;
>
> ---
>
> &nbsp;
>
> **Q4.** Yes. Our musculoskeletal model (169 cm, 72.9 kg) falls within the demographic range reported in GEMS [2]: height: 173 ± 8.2 cm, weight: 68.4 ± 6.3 kg,  n=5.
>
> &nbsp;
>
> ---
>
> &nbsp;
>
> **Q5.** Although GP-based BO can converge with approximately 100 samples for a single 2D problem, our setting requires repeatedly solving such problems because step frequency, step length, and walking speed vary continuously across gait and tasks. Each simulation run requires approximately 5 seconds per parameter, implying that BO-GP would require roughly 10 minutes per optimization, which is impractical for real-time adaptation. In contrast, our surrogate enables real-time gradient-based optimization, which is essential for online use.
>
> Regarding the 60k samples, they span the entire 4D search space, whereas for a single gait parameter set, only about 200 parameter evaluations are required. Even a coarse grid with 15 points per dimension already yields around 50k samples. Given the strong nonlinearity and sharp transitions present in assisted gait dynamics, this sampling density is required to accurately capture the landscape. For these reasons, we believe that using 60k samples is justified.
>
> &nbsp;
>
> ---
>
> &nbsp;
>
>
> **References**
>
> [1] Park, Jungnam, et al. "Generative gaitnet." ACM SIGGRAPH 2022 Conference Proceedings. 2022.
>
> [2] Lim, Bokman, et al. "Delayed output feedback control for gait assistance and resistance using a robotic exoskeleton." IEEE Robotics and Automation Letters 4.4 (2019): 3521-3528.

---

> ### Comment · Reviewer_BgQH · 2025-11-15
>
> I thank the authors for the quick and detailed reply, which clarifies many of my confused points. I have some following-up questions:
>
> 1. During the gait generator training, are the control parameters randomly sampled and fed to the gait generator? Is there large difference of the resulting gait when applying different control parameters?
>
> 2. During the loop of evalPWS, are the control parameters also random sampled instead of fixed when evaluating CoT (which is not shown in the pseudocode)? If so, how did you aggregate different CoT under different control parameters?
>
> 3. Given 5 seconds per CoT evaluation, does the 60k data collection of Algorithm 3 cost about 300k seconds (3.5 days)?
>
> 4. Would the gradient-based optimization over the surrogate model stuck in some local optima? Given 10 minutes of BO optimization is not a long time, why real-time is a requirement in practice?

---

> > ### Author Response · Authors · 2025-11-15
> >
> > Thank you for your continued feedback and for the follow-up questions.
> >
> > &nbsp;
> >
> > ---
> >
> > &nbsp;
> >
> > **Ans1.** Yes, during training the control parameters are randomly sampled.
> > We observed that different control parameters lead to different kinematics as well as CoT in the simulation.
> >
> > &nbsp;
> >
> > ---
> >
> > &nbsp;
> >
> > **Ans2.** The CoT curve is typically measured from human walking without an exoskeleton. Accordingly, the trainDataGenerator in Algorithm 1 is trained without using any exoskeleton control parameters. We will clarify this more explicitly in the revision.
> >
> > &nbsp;
> >
> > ---
> >
> > &nbsp;
> >
> > **Ans3.** The operations in Algorithm 3 (Lines 2–4) can be executed in parallel with different control parameter sets. The full data collection approximately took less than two hours in a moderate workstation (60k sample, 128 core).
> >
> > &nbsp;
> >
> > ---
> >
> > &nbsp;
> >
> > **Ans4.** Like other optimization methods for nonlinear and noisy objectives, our approach can converge to sub-optimal solutions depending on factors such as initialization or random seed. However, incorporating the gradient-penalty term into the surrogate loss reduces sensitivity to noisy samples and yields a smoother optimization landscape.
> >
> > With our optimization method, the optimal parameters can be identified as soon as a user wears the exoskeleton, and the system can adapt online to changes in the user's gait parameters, which commonly occur in real scenarios. In contrast, if optimization requires several minutes as in the case of BO, such immediate use becomes infeasible. Online adaptation would only be possible if the optimizations for all possible gait parameters had been precomputed in advance and stored on the device.

---

> > > ### Comment · Reviewer_BgQH · 2025-11-16
> > >
> > > Thank for your reply. I have no further questions. I think it would be an interesting work if incorporating the above clarifications in the main paper, and I look forward to more results about employing Exo-plore to real-world exoskeleton optimization applications. I have raised my score to 8 (accept).

---

### Official Review · Reviewer_xhMH · 2025-10-29

**Soundness:** 4
**Presentation:** 4
**Contribution:** 3
**Rating:** 10
**Confidence:** 4

**Summary:**

The paper presents a very advanced simulative biomechanical model, called Exo-plore, to study lower limb gait kinematics, kinetics, and muscular activity. Exo-plore allows to simulate also wearable assistive devices such as a hip assistive exoskeleton integrated with the biomechanical model. This allows to study more advanced control strategies for wearable systems. The simulation framework is benchmarked against actual human experimental data showing promising result. The work is novel, well presented, and relevant.
I would greatly appreciate a better quantification of the differences between simulated and experimental results and a more insightful evaluation of such discrepancies. I honestly believe those difference could elucidate not only the simulative environment limitations but also show some potentially interesting biomechanical features of motor control.

**Strengths:**

1. The Exo-plore simulation-based framework represents a notable step forward in the development of biomechanical simulation environments that reduce the sim2real gap. The framework demonstrated promising results when benchmarked against experimental data despite the complexity of the musculoskeletal system.

2. The possibility to simulate assistive devices in combination with biomechanical models is also another promising result. Fine-tuning exoskeletal control parameters is often hard, and Exo-plore might help developing faster and more reliable controllers.

2. I find convincing the definition of the total reward, and I especially appreciate the level of detail and reasoning behind it. Often times, papers neglect the proper presentation of the rationale behind the reward function. However, reward shaping is fundamental and has a very-well known impact on the performance of learning algorithms.

3. The idea of the resistance-minimization hypothesis is compelling and somehow novel in this context. I found this a strength of the proposed work, which differentiates it from the state-of-the-art.

4, The paper is clearly presented, and it provides a significant step forward in the development of realistic lower-limb musculoskeletal models capable to reproduce multiple gait patterns and different assistive scenarios.

**Weaknesses:**

1. Authors could improve the literature on human-in-the-loop optimization for exoskeletons. The group of Prof. Elliott Rouse at the U. Michigan has done extensive work on the topic on both prostheses and exoskeletons, but I could only find 1 work from his group in your references. Consider extending your literature review and include also this relevant portion of literature.

2. In neuromechanical simulations, authors should briefly mention the fact that Hill-type muscle actuators are an approximation of the actual muscle behavior. More specifically, Hill never designed the muscle model to be used for complex musculoskeletal models but only to investigate the force-speed relationship in human individual muscles. As a consequence, many other features of muscle dynamics are not captured by it. Nonetheless, most musculoskeletal models employ Hill-type muscle models ''carelessly'' believing that it gives a complete representation of human behavior. This aspect can lead users of such models to incorrect insights on the actual human musculoskeletal system.
Please consider mentioning this aspect in your digression.

3. It would be nice if you could provide some measure of similarity (or distance) between the simulated and experimental behaviors - both at the joint and muscular level, as well as for the other results presented in the paper. For join trajectories, I agree that the hip internal rotation is the one presenting the largest difference, while the other joint angular trajectories are closer to the behavior of human subjects. Nonetheless, some differences might be physiologically relevant. For example, in Figure 2 - Knee Flexion panel, you can observe that the simulation generates: (i) an overall smaller knee flexion peak during swing, and (ii) a practically negligible knee flexion during stance. Why is that? The knee flexion peak during swing is fundamental in normal walking to reduce the risk of stumbling and potentially falling, while the knee flexion peak during stance is important for shock absorption purposes. Is there any insight on why the model is not able to capture these features? Analogous observations could be done for the ankle dorsiflexion or hip abduction. I would greatly appreciate if you could comment more extensively on these results. (Similar comments are reported in the Questions also for Figure 5).

**Questions:**

1. Is metabolic rate efficiency i.e., energy efficiency, the 'holy grail' of gait? Most of the existing research focuses on developing lower-limb exoskeletons that lower metabolic cost. However, humans do not always plan for most efficient behavior. It might be worth discussing this aspect in the paper. I also recommend having a look at this interesting (and recent paper): https://www.nature.com/articles/s44172-023-00091-2

2. Equation (1) is clear but I struggle to understand why you apply a torque proportional to the difference in sine of the hip angles? Why not applying a torque proportional to the actual difference between the angles? What's the need of the sine function? Moreover, 'k' has the dimensions of a stiffness, thus making your controller equivalent to an impedance control system. Is that correct? If so, I would mention it.
Last but not least, delayed-feedback control can become unstable. Have you considered this aspect?

3. While I understand the need to have an overall goal for the optimizer, why do most papers focus on metabolic cost minimization? Is this the actual human underlying goal during walking/gait? I think this is a non-trivial question. I don't expect you to answer quantitatively to this point - but it might be beneficial to specify somewhere that the metabolic cost minimization is only one of the possible goals of walking/gait, but other goals might exist.

4. In Figure 4, the step frequency and step length results are also interesting. I invite the authors to compute any form of error, similarity or distance metric between experimental and simulated results. I agree the similarity is good, but I also wonder what the discrepancies are telling us. For example, we can observe that the simulations present the best fit at 4 km/h, while they tend to diverge at the extremes (either higher or lower). Why is that? Is this a bias of the training or is this an indicator that some other aspects of the musculoskeletal system need to be included? (Also, but a minor point, the font of the plot labels (in most figures) is very small - can you slightly increase it?)

5. In page 7, line 377, you write (k,\Delta_t) = (8 Nm, 0.25s). But wouldn't 'k' represent physically a stiffness i.e., its unit of measurement shouldn't it be Nm/rad?

6. You write that the overall trends in Figure 5 match reasonably well between simulation and human subjects, and this is true especially for the exo assist moment. However, if we look at the hip flexion angle we observe that the simulation does not capture the fact that at the increase of the walking speed the hip flexion angle presents a left-shift. Why isn't the simulation capturing this? Is this something the simulation can or cannot capture? What is the source of this dissimilarity?
Moreover, the overall exo assist power is underestimated (in the simulation) by almost a factor of 4 or even 5 at times. Why is this happening? Furthermore, why did you not compute any form of similarity metric between experimental and simulated results?

7. Lines 388-389, there are two small typos: (i) "an downward-opening" should be "a downward-opening", (ii) "a upward-opening" should be "an upward-opening".

8. Line 978-979. Can you clarify why do you need to add a minimum mass constraint to stabilize the simulation? This aspect is unclear.

9. Table I.4. Please see my previous comment on the unit of measurement of ‘k’. Shouldn’t that be a stiffness? If not, please explain why.

10. Line 1121-1122. I believe there is a typo in the title of section M. Shouldn’t it be “Reduction” and not “Recution”?

---

> ### Author Response · Authors · 2025-11-15
>
> Thank you for your comments and helpful suggestions. We appreciate the feedback on the revisions. Please let us know if any points require further clarification or additional information.
>
> &nbsp;
>
> ---
>
> &nbsp;
>
> **W1.** We appreciate the suggestion regarding Prof. Elliott Rouse’s group and will expand the literature review in the revision.
>
> &nbsp;
>
> ---
>
> &nbsp;
>
>
> **W2.** We agree that Hill-type muscle models have inherent limitations. As noted, they are derived under quasi-static assumptions and do not capture several physiological properties such as tension transients after rapid length changes [1]. We will include this discussion in the revision to avoid overstating model fidelity.
>
>
> &nbsp;
>
> ---
>
> &nbsp;
>
> **W3.** We will include similarity metrics (e.g., DTW) for joint trajectories in the revision. The reduced knee/ankle flexion arises because the RL policy is trained in a disturbance-free environment, resulting in stiffer but more, energy-saving gait patterns. We will add a discussion on this behavior in the revision.
>
> &nbsp;
>
> ---
>
> &nbsp;
>
>
> **Q1 + Q3.** We agree that metabolic minimization is not the sole objective of human gait. Factors such as comfort while wearing the device, fatigue during prolonged use, and even visual appearance may play important roles in exoskeleton controller design. Although we plan to examine these aspects in future work, the present study follows prior literature in adopting metabolic cost minimization as the primary objective. We will add a discussion of this point and cite the paper suggested by the reviewer.
>
> &nbsp;
>
> ---
>
> &nbsp;
>
>
> **Q2.** The sine function normalizes the hip angle to the range from [-1, 1], which is a design choice made by the manufacturer of the GEMS. The parameter k can indeed be interpreted as an impedance term, and we will clarify this in the revision. Stability issues for delayed-feedback control have been analyzed in prior work [2].
>
> &nbsp;
>
> ---
>
> &nbsp;
>
>
> **Q4.** We will compute Pearson correlation metrics in the revision. As noted in lines 358–361, foot model limitations likely contribute to the sim-to-real gap, particularly at extreme walking speeds. We will also enlarge the font sizes in all figures.
>
> &nbsp;
>
> ---
>
> &nbsp;
>
>
> **Q5 + Q9.** Because of the sine-based normalization, torque units were used; however, interpreting k as stiffness is clearer. We will revise the units accordingly and update Table I.4.
>
>
> &nbsp;
>
> ---
>
> &nbsp;
>
> **Q6.** The left-shift in hip flexion angle reflects an increased stance ratio at low walking speeds. Human walkers typically increase double-support time because reduced dynamics make balance control more challenging at slow speeds [3]. In contrast, the RL policy can react instantaneously at each simulation step and therefore has much more stable motor control than humans. As a result, the learned gait does not require the same compensatory increase in stance time and maintains a stance ratio closer to its preferred walking speed.
>
> Regarding the assistive power, our simulation produces a smaller hip joint range of motion than human experiments, which reduces joint velocity and consequently power. The larger hip excursion observed in the GEMS data is not captured because our exoskeleton–human interaction model is currently too abstract. Improving this component is future work.
> As noted in W3, we will include similarity metrics between simulated and experimental trajectories in the revision
>
> &nbsp;
>
> ---
>
> &nbsp;
>
>
> **Q7 + Q10.** Thank you for pointing out the typos. We will correct them in the revision.
>
>
> &nbsp;
>
> ---
>
> &nbsp;
>
> **Q8.** In physics simulation, excessively small link masses can cause numerical instability when solving Newton’s equations, making the system nearly singular. To prevent this issue, we assigned a minimum mass of 50g to each exoskeleton component, which results in a small deviation from the nominal device mass. We will clarify this explanation in the revision.
>
> &nbsp;
>
> ---
>
> &nbsp;
>
> **References**
>
> [1] Abbott, B. C., and X. M. Aubert. "The force exerted by active striated muscle during and after change of length." The Journal of physiology 117.1 (1952): 77.
>
> [2] Lim, Bokman, et al. "Delayed output feedback control for gait assistance and resistance using a robotic exoskeleton." IEEE Robotics and Automation Letters 4.4 (2019): 3521-3528.
>
> [3] Wu, Amy R., et al. "Mechanics of very slow human walking." Scientific reports 9.1 (2019): 18079.

---

### Official Review · Reviewer_MxFv · 2025-10-31

**Soundness:** 4
**Presentation:** 4
**Contribution:** 4
**Rating:** 8
**Confidence:** 5

**Summary:**

A neuro-mechanical simulation framework is presented which generates realistic gaits for not just able-bodied humans but also for certain pathological conditions. These gait simulations are generated using deep reinforcement learning with a novel reward term that captures the human-exoskeleton interaction. Further, a surrogate network trained from simulation data gathered from the gait generator, estimates a differentiable cost of transport. A gradient based optimizer then consumes the outputs of the surrogate network to compute optimal exoskeleton control parameters given certain gait parameters.

**Strengths:**

The paper is well written and organized. The problem is well motivated by the authors who position it well among existing literature. Bridging the gap between neuromuscular simulation and reality is a highly relevant open problem in robotics.

The paper also presents interesting formulations - The novel reward term r_{hei} for training the generative GaitNet, intra-muscular regularizer in the loss term to learn Muscle Coordination Network, and the Surrogate Network which estimates cost of transport from simulated data are technically sound ideas and the results and ablation studies around these terms show their benefit very clearly.

The paper presents convincing and thorough results across the board - the gait kinematics with and without assistance, optimal exoskeleton gain values for disabled individuals, The Cost of transport landscape are explained thoroughly. The ablation studies around the requirement for the r_{hei} reward term are also extremely useful for the reader to understand the necessity of adding this term to model human-exoskeleton interaction.

In my opinion one of the bigger strengths of the paper are a few insightful discussions presented in the paper that pose very valid questions about the gap in simulation and reality with neuromuscular systems and potentially inspire future research - A few examples are -
In figure 4, RL policy has difficulty producing short step lengths due to the rigid body dynamics and unwanted collisions.
In appendix K and figure K.6, the over activation of Rectus femoris because of reduced cost of transport compared to real humans.
In appendix N, The results indicate that effective exoskeleton prescription depends on not just pathological severity but also the underlying trade off between stability and energy specific to each biomechanical adaptation is a great insight.

**Weaknesses:**

As the authors themselves acknowledge, while the results here show promise in closing the sim-2-real gap in neuro-mechanical simulation, the ultimate test is how well this actually works in real world testing, which is still missing. So, a question still remains as to how effective the proposed methodology is? But this does not take away the good work and its contributions in this paper.

Small observations -
In equation 16, it's not clear what M and n stand here. There are a lot of variables described in this paper, it would be great to have a table that summarizes every variable.

In appendix A, It's not clear where this M_{ref} is obtained from. Also, the policy architecture optimized using PPO is not mentioned anywhere in the paper. It might actually be useful to include learning curves and details of the PPO training  in the appendix for the benefit of anyone trying to reproduce this paper.

**Questions:**

1) It's not clear how accurate the gait patterns are for pathological conditions as they are compared with normal gaits (figure J5) and not with real human data with the same pathological condition, is the issue here the lack of biomechanical data?

2) In figure 2, comparison of joint kinematics in simulation - How does the anthropomorphic data such as height and weight of the model (169 cm, 72.9 kg) compare with the real human data?

3) In the videos, the arm motion does not look like natural human motion at all, how much impact does that have on lower-body kinetics?

4) Lastly, I’m curious about the ground reaction forces in a gait cycle. Do they match biomechanics data? These might also be a great indication for how close these simulations are compared to reality. The contact forces would depend on how the contact dynamics is modeled, which might also explain some observations in the gait kinematics.

---

> ### Author Response · Authors · 2025-11-15
>
> Thank you for your comments and helpful suggestions. We appreciate the feedback on the revisions. Please let us know if any points require further clarification or additional information.
>
> &nbsp;
>
> ---
>
> &nbsp;
>
> **W1.** We agree with the reviewer’s point. We also plan to evaluate the effectiveness of our method in real-world settings in future work.
>
> &nbsp;
>
> ---
>
> &nbsp;
>
> **W2.** In the revision, we will clarify the definitions of M and n in Equation (16) and add a nomenclature table in the appendix..
>
> &nbsp;
>
> ---
>
> &nbsp;
>
> **W3.** The reference motion is adopted from the Generative GaitNet [1]. We will add an overview of the policy architecture and include the training curves in the appendix in the revision.
>
> &nbsp;
>
> ---
>
> &nbsp;
>
> **Q1.** Direct comparison with real pathological gait is challenging due to limited data availability and substantial inter-individual and inter-trial variability. Moreover, patients often exhibit compounded symptoms, which further complicates one-to-one comparisons. We will clarify this limitation in the revised manuscript.
>
> &nbsp;
>
> ---
>
> &nbsp;
>
> **Q2.** Gait120 dataset statistics [2] (173.2 ± 5.6 cm, 74.1 ± 12.5 kg) show that our model’s parameters (169 cm, 72.9 kg) fall within natural variation.
>
> &nbsp;
>
> ---
>
> &nbsp;
>
> **Q3.** The unnatural arm motion in the GIF results from truncated playback. The full video shows natural motion and minimal impact on lower-body kinetics [3].
>
> &nbsp;
>
> ---
>
> &nbsp;
>
> **Q4.** Here we provide the comparison between simulated and experimental ground reaction forces (GRFs) [4], using real data from [5]. We will include this plot and discussion in the revision.
>
> &nbsp;
>
> ---
>
> &nbsp;
>
> **References**
>
> [1] https://github.com/namjohn10/GenerativeGaitNet/blob/main/data/motion/walk.bvh
>
> [2] Boo, Junyo, et al. “Comprehensive human locomotion and electromyography dataset: Gait120.” Scientific Data 12.1 (2025): 1023.
>
> [3] https://drive.google.com/drive/folders/1u3a8VM24e_wfZ7WfxjJF8W1M_qOxRoZN?usp=sharing
>
> [4] https://drive.google.com/file/d/1PRpfnRegAfI1ucC1cM35lfFMQ_oB7mDL/view?usp=sharing
>
> [5] Horst, Fabian, et al. "Gutenberg Gait Database, a ground reaction force database of level overground walking in healthy individuals." Scientific data 8.1 (2021): 232.

---

> > ### Comment · Reviewer_MxFv · 2025-11-17
> >
> > Thanks for clarifications and additional information.
> >
> > I have a question about the GRF plot. DART resolves contact forces by formulating it as implicit LCP and solving resulting optimization problem - these usually lead to non-smooth GRF's with peaks at the first contact. Was any smoothing used in generating the GRF plots, or was it an entirely different contact solver? Actually, was any post processing done on any of the joint kinematics plots as well?
> >
> > The reason I am curious is this brings additional insight into how much of a sim-2-real gap is actually present.

---

> > > ### Author Response · Authors · 2025-11-18
> > >
> > > Thank you for your continued feedback and for the follow-up questions.
> > >
> > > **Ans.** For the GRF plots, we applied light post-processing to reduce solver-induced jitter in the contact forces. In contrast, the kinematics required no post-processing at all; we simply averaged results across multiple random seeds (n=11) for statistical reporting. Additionally, we believe that analyzing how GRF characteristics change under different contact solvers and how such differences may influence the resulting motion would be an interesting direction for future work.

---

### Official Review · Reviewer_9K5t · 2025-11-01

**Soundness:** 2
**Presentation:** 3
**Contribution:** 2
**Rating:** 4
**Confidence:** 3

**Summary:**

The paper introduces a neuromechanical simulation framework that couples a Deep RL–driven gait generator with a stochasticity‑aware surrogate optimizer to select hip‑exoskeleton control parameters, specifically a gain without human-in-the-loop experiments. Core ingredients are: (i) a gait data generator (PoseNet + PD control + a Muscle Coordination Network) trained with a composite reward that blends energy minimization and a human-exoskeleton interaction (HEI) term; and (ii) a surrogate network trained on simulated datasets.

**Strengths:**

- Clear problem framing and a convincing argument for simulation‑first controller selection when HILO is impractical. (Intro; Fig. 1.)
- Well engineered pipeline (PoseNet/MCN/IMR; LHS + Huber + gradient penalty) with thoughtful ablations and appendices documenting design choices. (Method; Appendix B, D-G).
- Clinically relevant analysis of pathological gaits

**Weaknesses:**

- The paper’s scope seems more suitable to a robotics or rehabilitation venue than here. While applications to a particular domain, such as, rehabilitation and robotics are a perfect fit for this venue, the core of this work lies primarily in the exoskeleton control and simulation framework rather than in a methodological advance in/applying deep learning itself.

- Fitting (α,β) to match the human CoT-speed parabola incorporates experimental behavior into the simulator (Algorithm 1–2). This aids realism but complicates claims of prediction vs post‑hoc alignment.

- The HEI term is explicitly engineered to reproduce delay/power scaling observed in data (Fig. 6–7). This is practical, but it risks a form of “trend imitation.” Evidence that other experimentally reported adaptations (not used in shaping) also emerge would clarify generality.

- The surrogate approach is compared mainly to GP‑capacity limits (Appendix D). Modern scalable GP approximations (e.g., inducing points) or kernel ridge with random features would be useful baselines.

- I missed comparison to other controllers ( e.g., the 2‑parameter delayed‑feedback controller to a phase‑based profile with 3–5 parameters)

**Questions:**

- How sensitive are optimized (κ,Δt) to the HEI scaling in Eq. (6)? Would normalizing by peak positive device power or by COM mechanical work be more robust across gains?

- Did you test parameterizations where Δt is specified in phase (as a fraction of gait cycle) rather than time, to reduce speed dependence and does the optimizer still recover the same pattern seen in Fig. 8d?

For more questions, please see the weakness section

---

> ### Author Response · Authors · 2025-11-15
>
> Thank you for your comments and helpful suggestions. We appreciate the feedback on the revisions. Please let us know if any points require further clarification or additional information.
>
> &nbsp;
>
> ----
>
> &nbsp;
>
> **W1.** We appreciate the reviewer’s understanding of our contributions, but we respectfully disagree with the claim that ICLR is not an appropriate venue.
> 1. As stated on the ICLR website, the conference explicitly welcomes work in “important application areas such as … computational biology … robotics,” which directly covers our domain.
> 2. RL-based neuromechanical control remains under-explored in robotics/rehabilitation venues. Presenting this work at an ML venue is therefore appropriate and valuable, as it helps broaden the scope of machine learning research toward scientifically important domains.
> 3. Our work also introduces methodological contributions: Algorithm 1 provides a computational method for integrating human metabolic objectives into an RL, and our surrogate optimization introduces a gradient-penalty term that significantly improves stability.
>
> &nbsp;
>
> ----
>
> &nbsp;
>
> **W2.** Optimization or learning over complex dynamical systems often falls into sub-optimal solutions. To mitigate this and improve the realism of our simulation results, we incorporate a fitting process for the parameters (α,β) based on the well-established relationships in gait. Developing a fully predictive system that does not require such fitting would be an interesting direction for future research.
>
> &nbsp;
>
> ----
>
> &nbsp;
>
> **W3.** The HEI term in Eq. (6) is intentionally kept as simple as possible, making it unlikely to capture the complex delay- and power-dependent patterns observed in experiments. Moreover, several additional results (Fig. 5, Fig. 7, Fig. M.7) show that these complex adaptation behaviors emerged naturally without explicit engineering. To the best of our knowledge, this work is the first to demonstrate human–exo adaptation using a RL-based neuromechanical simulation. Checking whether the proposed principle extends to other reported adaptations remains future work.
>
> &nbsp;
>
> ----
>
> &nbsp;
>
> **W4.** We initially tested standard GPs as surrogates, but their performance was insufficient for our highly nonlinear 4-D control space. We also evaluated a modern inducing-point method (KISS-GP/SKI via gpytorch [1]). In our setting, memory usage and performance were similar to vanilla GP regression, providing no practical advantage.
>
> &nbsp;
>
> ----
>
> &nbsp;
>
> **W5.** We targeted the GEMS device because human experimental data are available for its two-parameter controller. For future work, we are planning to extend Exo-plore to support a broader family of controllers.
>
> &nbsp;
>
> ----
>
> &nbsp;
>
> **Q1.** We will check the sensitivity of optimized (κ, Δt) to the HEI scaling and address this in the revision.
> Regarding the suggestion to normalize by peak device power or COM mechanical work, these quantities depend directly on the agent’s evolving gait and therefore vary during learning. Using such internally coupled signals for normalization can destabilize training. For this reason, we chose a constant gain (κ).
>
> &nbsp;
>
> ----
>
> &nbsp;
>
> **Q2.** Thank you for the suggestion. We compared the time-based representation with the phase-based representation, and the initial results indicate that the two measures do not differ significantly. We will include this finding in the revision after conducting additional trials for final confirmation.
>
> &nbsp;
>
> ----
>
> &nbsp;
>
> **References**
>
> [1] https://github.com/cornellius-gp/gpytorch

---

### Meta-Review · Area_Chair_xgrk · 2026-01-07

**Summary:**

This paper received a mixed set of ratings, with strong accept, accept, borderline reject and reject. All reviewers recognized the significance of the “simulation-first” approach, which directly addresses the impracticality of human-in-the-loop (HILO) experiments for mobility-impaired populations. This design aligns closely with clinical and robotics requirements, and the technical pipeline of the work was deemed well-structured.

That said, reviewers raised several common concerns:
- The absence of real-world validation was highlighted by all reviewers;
- Certain citations, definitions, and annotations were deemed ambiguous or inadequately clarified;
- The need for supplementary real-world experiments was emphasized.

The majority of these concerns have been resolved through the authors’ formal responses and revisions.

**Reviewer Concerns:**

All reviewer concerns have been resolved, save for those related to the critical sim2real challenge of translating simulation-based findings to practical, real-world applications.

**Reviewer Scores:**

Reviewer BgQH has raised the score from 2 to 8.

---

### Decision · Program_Chairs · 2026-01-26

Accept (Poster)